# Zone Method meets Physics-Informed Neural Networks: Data and Regularization for High-Temperature Processes

## Abstract

In this paper, we visit the classical Hottel's zone method from a Machine Learning (ML) perspective. The zone method provides an elegant, iterative way to model high temperature processes, such as in industrial furnaces, which are energy-intensive components in the Foundation Industries (FIs). Real-time and accurate temperature modeling in reheating furnaces can help the FIs achieve their Net-Zero goals for the greater good of environmental sustainability. As computational surrogate models are expensive, and slower, we propose to leverage Deep Learning (DL) owing to their effectiveness and inference efficiency. However, obtaining quality large-scale data for DL/ML training in systems with high temperature processes such as furnaces is challenging. To address this, we propose an algorithm to use the zone method for generating data for training ML/DL models via regression. But, DL inherently finds it challenging to generalize to Out-of-Distribution (OOD) samples. Thus, we additionally propose to employ zone method inspired novel Energy-Balance (EB) based regularizers to explicitly convert a neural network to a Physics-Informed Neural Network (PINN) for adhering to the underlying physical phenomenon during network training, to be able to better handle OOD data. We support our claims with extensive empirical experiments, thus showcasing the promise of our PINN. Our PINN framework has been demonstrated for both Multi-Layer Perceptron (MLP) and Long Short-Term Memory (LSTM) based Recurrent Neural Network (RNN) architectures.

## 1 Introduction

Many high temperature processes (operating at above $700°C$) appear in a number of real-world applications, such as, chemical reactors (Feng & Han, 2012), solar energy applications (Muhich et al., 2016; Marti et al., 2015), pebble bed nuclear reactors, infrared drying (Pawar & Pratape, 2017; Krishnamurthy et al., 2008), fluidized beds (Von Zedtwitz et al., 2007), selective laser sintering in 3D printing (Tran & Lo, 2018; Zhou et al., 2009), rotary kilns (Njeng et al., 2018; Emady et al., 2016; Yohannes et al., 2016), etc. A dominating heat transfer mechanism in such processes is Radiative Heat Transfer (RHT), which, in non-vacuum conditions naturally occurs among the other heat transfer mechanisms: conduction and convection. RHT is also prominently relied upon by thermal transport under vacuum conditions in astronomical applications.

The classical Hottel's zone method is a widely used technique for predicting RHT (Hottel & Cohen, 1958; Hottel & Saforim, 1967), especially in enclosures such as furnaces. In this paper, we visit the zone method from a Machine Learning (ML) perspective. Firstly, we propose an algorithm to generate a temporal data set using the zone method. Then, we recast the data into an Independent and Identically Distributed (IID) format so that any off-the-shelf ML model can be trained using this data, for predicting certain relevant output variables (via regression) pertaining to a real-world application involving high-temperature processes.

Due to the promise of Deep Learning (DL), we particularly assess the performance of a Multi-Layer Perceptron (MLP) neural network to model this task. However, DL inherently finds it challenging to generalize to Out-of-Distribution (OOD) samples. Thus, we additionally propose to employ zone method inspired novel Energy-Balance (EB) based regularizers to explicitly convert the MLP to a Physics-Informed Neural Network (PINN) for adhering to the underlying physical phenomenon during network training, to be able to better

handle OOD data. Our PINN framework, as we show in our experiments, can naturally encompass any generic regessor model requiring input-output pairs. We demonstrate this by also extending it to physics-informed Long Short-Term Memory (LSTM) based Recurrent Neural Network (RNN) architectures.

It should be noted that in a typical ML setting, we often do not have a clarity on the exact distribution from which a data set has been sampled from. On the other hand, in the field of scientific ML, since we do have a white-box computational model that has been used to simulate ML training data, the resulting ML models have a notion of *implicit physical awareness* by virtue of the (white-box) generated data. However, explicitly making the models physics-aware by virtue of regularization is a widely studied, and beneficial design construct in the PINN literature (Karniadakis et al., 2021). Along with the theoretical motivation, we support our claims by extensive empirical evaluation.

**Motivation:** Yuen & Takara (1997) in their study, have proved the elegance and superiority of the zone method over contemporary counterparts to model the physical phenomenon in high-temperature processes. In our work, we use the zone method towards a real-world application for the Foundation Industries (FIs), applied to reheating furnaces, due to the close and natural association/ relation of the zone-method with the latter. Foundation Industries (FIs) constitute glass, metals, cement, ceramics, bulk chemicals, paper, steel, etc. and provide crucial, foundational materials for a diverse set of economically relevant industries: automobiles, machinery, construction, household appliances, chemicals, etc. FIs are heavy revenue and employment drivers, for instance, FIs in the United Kingdom (UK) economy are worth £52B (EPSRC report), employ 0.25 million people, and comprise over 7000 businesses (IOM3 report). The rapid acceleration in urbanization and industrialization over the decades has also led to improved building design and construction techniques. Great emphasis has been gradually placed on efficient heat generation, distribution, reduction, and optimized material usage.

However, despite their economic significance, as depicted by the above statistics, the FIs leverage energy-intensive methods. This makes FIs major industrial polluters and the largest consumers of natural resources across the globe. For example, in the UK, they produce 28 million tonnes of materials per year, and generate 10% of the entire UK's $CO_2$ emissions (EPSRC report; IOM3 report). Similarly, in China, the steel industry accounted for 15% of the total energy consumption, and 15.4% of the total $CO_2$ emissions (Zhang et al., 2018; Liang et al., 2020). These numbers put a challenge for the FIs in meeting our commitment to reduce net Green-House Gas (GHG) emissions, globally.

Various approaches have been relied upon to achieve the Net-Zero trajectory in FIs (Net Zero by 2050): switching of grids to low carbon alternatives via green electricity, sustainable bio-fuel, and hydrogen sources, Carbon Capture and Storage (CCS), material reuse and recycling, etc. However, among all transformation enablers, a more proactive way to address the current challenges would be to tackle the core issue of process efficiency, via digitization, computer-integrated manufacturing, and control systems. Areas of impact by digitization could be reducing plant downtime, material and energy savings, resource efficiency, and industrial symbiosis, to name a few. Various computer-aided studies have already been conducted in notable industrial scenarios. The NSG Group's Pilkington UK Limited explored a sensor-driven Machine Learning (ML) model for product quality variation prediction (up to 72h), to reduce $CO_2$ emission by 30% till 2030 (IOM3 report). Similar studies on service-oriented enterprise solutions for the steel industry have also been done recently in China (Qin et al., 2022).

In this work, we tackle the key challenge of accurate and real-time temperature prediction in reheating furnaces, which are the energy-intensive bottlenecks common across the FIs. To give a perspective to the reader on why this is important, considering any process industry, such as the steel industry, one can observe that at the core, lies the process of conversion of materials (e.g., iron) into final products. This is done using a series of unit processes (Yu et al., 2007). The production process involves key steps such as dressing, sintering, smelting, casting, rolling, etc. A nice illustration of the different stages and processes in the steel industry can be found in Qin et al. (2022). The equipment in such process industries operates in high-intensity environments (e.g., high temperature), and has bottleneck components such as reheating furnaces, which require complex restart processes post-failure. This causes additional labor costs and energy consumption. Thus, for sustainable manufacturing, it is important to monitor the operating status of the furnaces.

A few studies (Hu et al., 2019) have shown promise in achieving notable fuel consumption reduction by reducing the overall heating time by even as less as 13 minutes while employing alternate combustion fuels. A key area of improvement for furnace operating status monitoring lies in leveraging efficient computational temperature control mechanisms within them. This is because energy consumption per kilogram of $CO_2$ could be reduced by a reduction in overall heating time.

However, available computational surrogate models based on Computational Fluid Dynamics (CFD) (Wehinger, 2019; De Beer et al., 2017), Discrete Element Method (DEM) (Emady et al., 2016), CFD-DEM hybrids (Oschmann & Kruggel-Emden, 2018), Two Fluid Models (TFM) (Marti et al., 2015), etc. incur expensive and time-consuming data acquisition, design, optimization, and high inference times. To break through the predictive capability bottlenecks of these surrogate models, DL approaches can be suitable candidates for real-time prediction, owing to their accuracy and inherently faster inference times (often only in the order of milliseconds).

As only a handful of sensors/ thermo-couples could be physically placed within real-world furnaces (and that too at specific furnace walls), the challenge of obtaining good-quality real-world data at scale in such scenarios remains infeasible. To alleviate this, we identify the classical Hottel's zone method (Hottel & Cohen, 1958; Hottel & Saforim, 1967) which provides an elegant, iterative way to computationally model the temperature profile within a furnace, requiring only a few initial entities which are easily measurable. However, straightforward utilization of the same is not suitable for real-time deployment and prediction, due to computational expensiveness.

**Our proposal:** To this end, we propose to leverage the zone method to rather generate a data set *offline*, and train ML models which can then be used for real-time prediction. From a ML perspective, **our contributions/ claims** are:

1. **(Data):** We discuss how the zone method could be used to first generate a temporal data set (with entities in a time step dependent on some of the entities from the previous time step), and then recast it into a set of Independent and Identically Distributed (IID) instances for training off-the-shelf ML models in a regression setting (i.e., given few inputs, predict certain output values, corresponding to each time step of furnace operation).

2. **(Regularization):** We further propose explicit regularization of a neural network by virtue of our novel, energy-balance based constraints, backed by the zone method. The resulting PINN is expected to generalize well to Out-Of-Distribution (OOD) samples. To showcase this OOD generalizability, during data set creation, we make sure that the distribution (in our case, a furnace configuration) of the test data are different and disjoint from that of the training data, for all our experimental settings. We later empirically demonstrate that compared to a naive neural network, our PINN in general performs better on the test data, thereby highlighting the OOD generalizability.

3. The data creation method discussed in our method is holistic, covers all possible exchange areas (as compared to related works, as discussed later), and thus, is unique in nature itself. We further discuss recasting this data from temporal nature to an IID one, which is a hallmark of our work. To evaluate on this data, it is only meaningful to start with simpler baselines from classical ML, and a simple baseline MLP without physics awareness. We gradually start with these simpler baselines in our experiments, and then introduce explicit physics based regularization within the MLP architecture.

   We compare the proposed PINN extensively against various classical ML and neural network based baselines, to highlight its effectiveness. In particular, we compare against Decision Trees, Random Forests, Boosting, and Multi-Layer Perceptron (MLP). The reason for choosing these baselines is due to the fact that the IID data can be represented in a tabular format, and for data of such form, these baselines are widely adopted in the industry due to their effectiveness despite their *simplicity*.

4. We also compare our PINN against a Long Short-Term Memory (LSTM) based Recurrent Neural Network (RNN) which is well-suited for modeling temporal data. We further showcase that our PINN framework being generic in nature, can also be used to obtain Physics-Based LSTM variants equipped with our energy-balance regularizers.

5. While a PINN performing better than a MLP or other baselines, is a well-studied concept, establishing the same for a newer problem, with a novel formulation, as done by our work, is a core contribution

of our paper. Additionally, our setup being a generic framework, is geometry-agnostic of the 3D structure of the underlying furnace, and as such could accommodate any standard ML regression model, requiring input-output pairs.

While the research conducted in this work is at nascent stage, we believe it could pave way for further developments from an ML perspective, to solve a real-world application problem with value in terms of environmental sustainability. Our work, for an applied physical sciences reader, could inspire how ML and DL could be used to address a niche domain scenario. At the same time, for an ML audience, we believe that our work showcases a novel way to integrate physics based constraints into a neural network, especially using the zone method. Arguably, there exists a plethora of works related to PINNs, however, using PINNs to incorporate the zone method based regularizers as in our work, is a novel contribution to the community. The motivation to leverage the zone method also comes from the fact that it provides an elegant (and superior) way, as studied by Yuen & Takara (1997), to model the physical phenomenon in high-temperature processes inside reheating furnaces.

**Paper organization:** In Section 1, we introduce our work, its motivations, and our proposed claims. The rest of the paper is organized as follows: In Section 2 we discuss a few related works, and mention the uniqueness of our paper with respect to the existing body of relevant literature. We discuss the proposed methodology in Section 3, after providing a relevant background. In Section 4, we present our key empirical findings and observations. We conclude the paper with necessary details of data set creation, experimental details and additional experiments, including an in-depth analysis of our method in the Appendix section A.

## 2 Related Work

In this section, we exhaustively present a set of relevant approaches with which our work can be loosely associated with. Specifically, we categorize them into two major classes: i) nonlinear dynamic systems, radiative heat transfer and view factor modeling, and, ii) modeling in reheating furnaces. We also talk about PINNs, and how our method is unique with respect to the existing literature.

**(Category 1) Nonlinear dynamic systems, radiative heat transfer and view factor modeling:** Our work at its heart is based on the zone method, which in turn relies on notions of radiative heat transfer and view factor modeling (or interchangeably, exchange area calculation). Describing the behavior of a furnace state involves combustion models, control loops, set point calculations, and fuel flux control in zones. It also involves linearization and model order reduction for state estimation and state-space control. The inherent complexity makes the modeling a nonlinear dynamic system.

While there is no exact similarity, our work shares some common philosophies with few earlier works. For instance, Ebrahimi et al. (2013) discuss the modeling of radiative heat transfer using simplified exchange area calculation. Radiative heat transfer in high-temperature thermal plasmas has been studied by Melot et al. (2011) while comparing two models. A nonlinear dynamic simulation and control based method has been studied by Hu et al. (2018). A classical work based on genetic algorithm for nonlinear dynamic systems (Li, 2005) is also present, which, instead of a data-driven approach, leverages a pre-defined set of mathematical functions.

Within this category, some approaches have also employed neural networks. In Yuen (2009), a network was trained for simulating non-gray radiative heat transfer effect in 3D gas-particle mixtures. Some approaches have used networks for view factor modeling with DEM-based simulations (Tausendschön & Radl, 2021), and some have addressed the near-field heat transfer or close regime (García-Esteban et al., 2021).

**(Category 2) Modeling in reheating furnaces:** We now discuss methods dealing with some form of prediction or optimization in reheating furnaces. Classically, Kim & Huh (2000) discussed a method to predict transient slab temperatures in a walking-beam furnace for rolling of steel slabs. Kim (2007) proposed a model for analyzing transient slab heating in a direct-fired walking beam furnace. Jang et al. (2010) investigated the slab heating characteristics with the formation and growth of scale. Tang et al. (2017) studied slab heating for process optimization. A distributed model predictive control approach was proposed in Nguyen et al. (2014). Few multi-objective optimization methods were discussed in Hu et al. (2017); Ban et al. (2023). A fuel supplies scheme based approach was proposed in Li et al. (2023). Other related works involved multi-mode

model predictive control approach for steel billets (Zanoli et al., 2023), and a hybrid model for billet tapping temperature prediction (Yu et al., 2022).

Some neural network based approaches in this category studied transfer learning (Zhai & Zhou, 2020; Zhai et al., 2023), digital twin modeling (Halme Ståhlberg, 2021), and steel slab temperature prediction (de Souza Lima et al., 2023). Liao et al. (2009) discussed an integrated hybrid-PSO and fuzzy-NN decoupling based solution. Other works have studied aspects related to time-series modeling (Hwang et al., 2019; Chen et al., 2022), and multivariate linear-regression in steel rolling (Bao et al., 2023).

**PINNs:** The methods mentioned above discuss alternatives aimed at modeling either exchange factors with radiative heat transfer, or specific slab temperature predictions in reheating furnaces. However, they do not explicitly address physics-based prior incorporation within their optimization frameworks, especially for the neural network variants. To this end, we now discuss a few relevant works in the body of literature on PINNs. For a detailed review on PINNs in general, we refer the interested reader to the paper by Karniadakis et al. (2021). It should be noted that PINNs are a broad category of approaches, and the literature is vast. Here, we discuss those methods which relate to certain aspects of thermal modeling.

Drgoňa et al. (2021) proposed a physics-constrained method to model multi-zone building thermal dynamics. A multi-loss consistency optimization PINN (Shen et al., 2023) was proposed for large-scale aluminium alloy workpieces. Other approaches focus on prototype heat transfer problems and power electronics applications Cai et al. (2021), minimum film boiling temperature (Kim et al., 2022), critical heat flux (Zhao et al., 2020), solving direct and inverse heat conduction problems of materials (He et al., 2021), lifelong learning in district heating systems (Boca de Giuli, 2023), PINN and point clouds for flat plate solar collector (Han et al., 2023), residential building MPC (Bünning et al., 2022), hybrid ML and PINN for Process Control and Optimization (Park, 2022), reinforcement learning for data center cooling control (Wang et al., 2023), flexibility identification in evaporative cooling (Lahariya et al., 2022), and fast full-field temperature prediction of indoor environment (Jing et al., 2023).

**Uniqueness of our work within existing literature:** While we have observed a number of loosely related methods as discussed above, upon a clear look at them, we can conclude the following:

1. **Comparison with category 1 methods:** Among the approaches focusing on view factor modeling with radiative transfer, the area of interest is often simplified. The modeling covers select few exchange areas. The methods are also geometry-specific. Our approach on the other hand seeks a generic, geometry-agnostic modeling that covers the entire set of exchange areas. The exchange areas can be intuitively perceived as those interfaces from where radiation can transfer, between a pair of zones (surface/gas). We will provide a background on exchange areas in the proposed work section.
   The ones involving neural networks, often employ feed-forward Multi-Layer Perceptron (MLP) models with few hidden layers. We later showcase in our experiments that a simple MLP trained to regress the outputs given certain inputs, may not generalize well to unseen distributions, due to lack of explicit understanding of the underlying physics. On the other hand, we empirically showcase that our proposed PINN performs better than such a baseline MLP. Within a single PINN framework, our method can also cover other architectures such as LSTMs.

2. **Comparison with category 2 methods:** Both non-neural and neural-network based methods presented in this category, as observed, focus on predicting temperatures only in certain regions of a furnace, often, the slab temperature profiling. Our work, on the other hand aims at achieving a complete furnace temperature profiling, ranging from the gas zones, to both types of surface zones: furnace walls as well as the slab/obstacle surfaces. Our training data set is obtained based on the iterative zone method, and is more holistic in nature as compared to the discussed methods. This makes an apple-to-apple comparison difficult with other methods as they deal with different problem setups. Furthermore, the neural methods in this category are not trained to be physics aware.

3. **Comparison with PINNs:** It should be noted that any PINN approach is driven by the priors corresponding to the underlying physical phenomenon. As we did not find PINN methods addressing zone method based modeling, we could claim our PINN variant to be novel in nature, especially, in this studied problem setup. Essentially, casting the temperature prediction task in reheating furnaces as a regression task, and modeling via explicit physics-constrained regularizers (based on

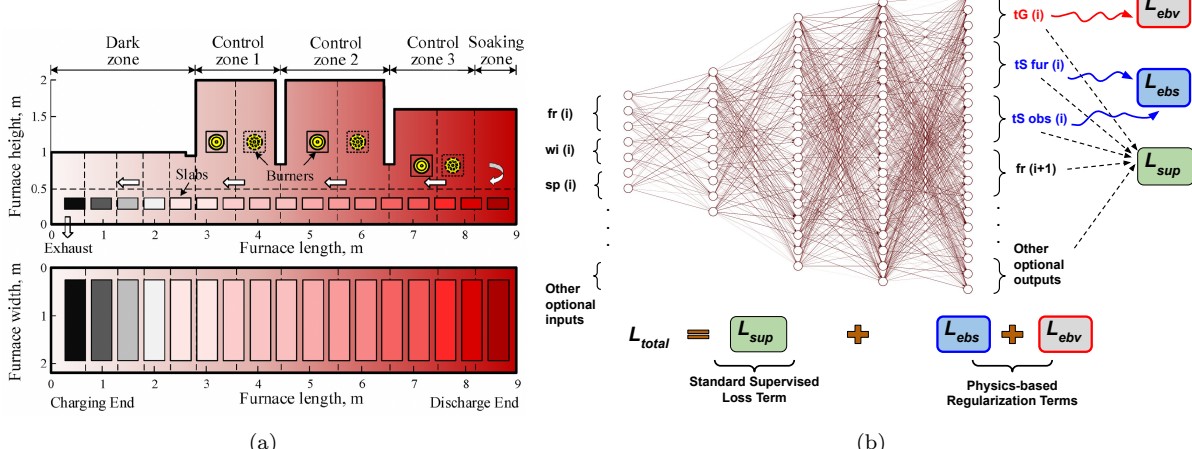

(a)                                                                 (b)

Figure 1: This figure is best viewed in color. Sub-figure (a) Illustration of a real-world furnace, and its subdivision as different zones. Image courtesy: (Hu et al., 2019). A darker shade of red indicates a higher temperature. Under normal conditions, temperature increases towards the discharge end. Sub-figure (b) Illustration of incorporating zone method based regularization to train a Physics-Informed Neural Network (PINN).

zone method) as done in our work, is a first of its kind. It is a simple paradigm, and could be used to build further sophisticated developments. At the same time, it simply requires input-output pairs (as shown later) to train the underlying ML/PINN model, and makes no geometry-specific assumptions of the furnace.

The data creation method discussed in our method is holistic, covers all possible exchange areas, and thus, is unique in nature itself. We further discuss recasting this data from temporal nature to an IID one, which is a hallmark of our work. To evaluate on this data, it is only meaningful to start with simpler baselines from classical ML, and a simple baseline MLP without physics awareness. We gradually start with these simpler baselines in our experiments, and then introduce explicit physics based regularization within the MLP, as well as LSTM architectures.

## 3 Proposed Method

### 3.1 Background

The zone method is widely used for predicting RHT in furnaces, by subdividing the furnace chamber into volume and surface zones. The volume zones correspond to the gaseous regions, and the surface zones, as implied by the name, correspond to the two types of surfaces: i) the furnace surface walls, and ii) surfaces of obstacles (slabs) to be heated. The temperature within a zone is assumed to be uniform. A set of simultaneous equations, called as the **Energy-Balance (EB)** equations govern the radiation exchange between the zones, which form the basis of our PINN model as explained later. These equations are representatives for the incident and leaving fluxes at each zone, and are updated at each iteration/step of the zone method to obtain the relevant temperature profiling.

To further represent the exchange of radiation between each pair of zones, we pre-compute a set of Total Exchange Areas (TEAs), which do not need to be modified during iterative updation of the EB equations. The TEAs need to be derived for every S-S, S-G, and G-G pair, where S and G respectively denote Surface and Gas/Volume. The radiant interchange between each zone pair is determined using the Directed Flux Areas (DFAs), which in turn depend on the TEA terms. Another advantage of the zone method is its ability to handle non-grey gases by using the Weighted Sum of Grey Gases (WSGG) model. In WSGG, combustion gases are represented by a mixture of grey gases plus a clear gas.

At the simplest level, a single volume zone surrounded by few surface zones can be used to represent a furnace, where, inter-zone radiation exchange can be ignored. In our study, the zone method is treated in a generic manner, to be applicable for complex geometries with 2-D/ 3-D arrangements of zones (e.g.,

Figure 1a). The iterative process can be solved with a few easily obtainable temperature-based entities and pre-computed TEA terms (explained along with our method). Please refer Appendix A for details on the following: Exchange Factors, TEA, DFA, and WSGG.

## 3.2 (Claim 1/ Data) Zone method based data generation for ML/ DL model training

We introduce our proposed method with the first claim of our paper. We identify the zone method as a suitable candidate for data generation intended to train DL/ML models for temperature prediction in reheating furnaces. We propose using it to generate an IID data set for training any off-the-shelf ML based regression model. In doing so, we also discuss 2 different ways/ settings to set our desired model inputs.

In a real-world furnace as in Figure 1a, the release of combustion materials (by burners, controlled via their firing rates) and movement of objects to be heated (slabs, or obstacles) from the left to the right (discharge end, with higher temperature), causes energy and mass flow. The zone method mathematically models this by dividing the furnace into a set of zones: i) G: Gas/ volume and ii) S: Surface (consisting of furnace walls *fur* and obstacle surfaces *obs*). The radiation interchange ($\leftharpoonup$ indicates the direction of flow) among all possible pairs $(i, j)$ of zones: Gas to Gas ($\overleftarrow{G_i G_j}$), Surface to Surface ($\overleftarrow{S_i S_j}$), Surface to Gas ($\overleftarrow{G_i S_j}$), and Gas to Surface ($\overleftarrow{S_i G_j}$), can be modeled along with a set of **Energy-Balance (EB) equations**.

Hu et al. (2016) has proposed a computational model of the zone method, which though highly accurate, is slower for real-time prediction. We use it, and simulate an offline, IID data set $\mathcal{X}_{IID} = \{(\boldsymbol{x}^{(i)}, \boldsymbol{y}^{(i)})\}_{i=1}^N$ for DL training, by following their algorithmic flow. The advantage of this model is that the required input entities (e.g., ambient temperatures, set point temperatures, firing rates, walk-interval) are readily available without dependency on the physical placement of sensors in every relevant location where we want to collect data in the real-world.

We study the following two settings:

1. **Input Setting 1 - Without previous temperatures in the input vector:** Here, for time step instance $i$, we set: $\boldsymbol{x}^{(i)} = [fr(i)^\top, wi(i)^\top, sp(i)^\top]^\top$, and $\boldsymbol{y}^{(i)} = [tG(i)^\top, tS\ fur(i)^\top, tS\ obs(i)^\top, fr(i+1)^\top]^\top$.
2. **Input Setting 2 - With previous temperatures in the input vector:** Here, for time step instance $i$, we set: $\boldsymbol{x}^{(i)} = [fr(i)^\top, wi(i)^\top, sp(i)^\top, tG(i-1)^\top, tS\ fur(i-1)^\top, tS\ obs(i-1)^\top]^\top$, and $\boldsymbol{y}^{(i)} = [tG(i)^\top, tS\ fur(i)^\top, tS\ obs(i)^\top, fr(i+1)^\top]^\top$.

Here, $fr(i)$, $wi(i)$, $sp(i)$, $tG(i)$, $tS\ fur(i)$ and $tS\ obs(i)$ are respectively the vectors containing firing rates, walk-interval, set point temperatures, gas zone temperatures, surface zone temperatures for furnace walls, and surface zone temperatures for obstacles for a time step $i$. Also, $tG(i-1)$, $tS\ fur(i-1)$, and $tS\ obs(i-1)$ are respective vectors containing the corresponding temperatures from the previous time step. $fr(i+1)$ is a vector containing firing rates for the next time step.

Using $\mathcal{X}_{IID} = \{(\boldsymbol{x}^{(i)}, \boldsymbol{y}^{(i)})\}_{i=1}^N$, we can estimate parameters $\theta$ of a neural network $f_\theta(.)$ by training it to predict $\boldsymbol{y}^{(i)}$ given $\boldsymbol{x}^{(i)}$, for all $i$, as:

$$\theta^* \leftarrow \arg\min_\theta \mathbb{E}_{(\boldsymbol{x}^{(i)}, \boldsymbol{y}^{(i)}) \in \mathcal{X}_{IID}} [||\boldsymbol{y}^{(i)} - f_\theta(\boldsymbol{x}^{(i)})||_2^2] \tag{1}$$

Then, we can obtain the required values of temperatures by extracting them from $f_{\theta^*}(\boldsymbol{x}^{(i)})$. Here, $f_\theta(.)$ can be any other standard ML model for regression. For neural network variants in our experiments, we study the Multi-Layer Perceptron (MLP) and Long Short-Term Memory (LSTM) based Recurrent Neural Network (RNN) architectures.

### 3.2.1 Temporal to IID data set creation

As the zone method is iterative in nature, entities in a time step are dependent on some of the entities from the previous time step. We now discuss how an inherently time dependent data set obtained by the zone method could be used for the IID data set generation. We use the zone method implementation provided by Hu et al. (2016), to represent various furnace configurations of the real-world furnace shown in Figure 1a. We

consider various different configurations, and create disjoint train-val-test splits in such a way that there is no overlap in the data across different splits. Also, each configuration could belong to either of train/val/test split. As val/test data belong to furnace configurations different from that of training, it naturally makes the test data OOD in nature. Each configuration can be uniquely defined by set point temperatures and the walk-interval. Set point temperatures are essentially the desired temperatures that the furnace is expected to achieve at different stages/ zones.

Note that we consider configurations with both normal conditions (increasing set points towards discharge end, as naturally occurring in practice), as well as abnormal ones (arbitrary set points). The details are present in Table 4 (in the Appendix). Here, each configuration contains a number of time steps. Within a configuration, each time step is sampled with a certain delay, to account for conduction analysis.

In Algorithm 1, we outline the key steps required in the data generation step, for a particular configuration. Please refer Hu et al. (2016) for further details and insights on the flow. Here, the major entities as discussed in our formulation are mentioned: $fr(t), wi(t), sp(t), tG(t), tS\ fur(t), tS\ obs(t)$. In addition, we also make use of auxiliary entities required by our PINN later, such as enthalpy $\boldsymbol{q}^{(t)}$, heat-flux $\boldsymbol{w}^{(t)}$, and node temperatures $\boldsymbol{n}^{(t)}$, all of which are vectors containing values from across zones.

The algorithm requires initial values of set points and walk interval to fix a furnace configuration, along with the number of steps $T > 0$ considered for data generation using that configuration. In practice, the initial set of ambient temperatures across various volume and surface zones will also be readily available with the knowledge of the furnace operator. The initial firing rates corresponding to each of the burners will also be known, for providing as inputs. Thereafter, for each time step, the new zone temperatures and firing rates could be obtained using the following iterative steps in the order of (a visual illustration of this data generation flow can be found in the Figure 2):

1. **Firing rates updation:** Using the previous values of temperatures, and firing rates, update the new firing rate.
2. **Flow pattern and Enthalpy computation:** The firing rates trigger a flow-pattern within the furnace, across (a mix of) volume and surface zones. This can in turn be used to compute the enthalpy terms in the furnace.
3. **DFA using TEA:** The pre-computed Total Exchange Area (TEA) terms $\boldsymbol{GG}, \boldsymbol{GS}, \boldsymbol{SG}, \boldsymbol{SS}$, along with temperatures from the previous time steps, are used to compute the Directed Flux Area (DFA) terms $\overleftarrow{\boldsymbol{GG}}^{(t)}, \overleftarrow{\boldsymbol{GS}}^{(t)}, \overleftarrow{\boldsymbol{SG}}^{(t)}, \overleftarrow{\boldsymbol{SS}}^{(t)}$.
4. **EBV equations:** The gas zone temperatures for the time step in consideration are then computed using the EBV equations (abbreviation explained later) involving the relevant gas zone related DFA terms and the enthalpy terms. These, along with the remaining surface related DFA terms and the previous surface temperatures are used to compute the heat fluxes $\boldsymbol{w}^{(t)}$.
5. **Conduction Analysis and EBS equations:** Conduction analysis is performed on $\boldsymbol{w}^{(t)}$ to obtain the node temperatures, which are then used in the EBS equations (abbreviation explained later) to obtain the updated surface zone temperatures for the current time step.
6. Collect all relevant entities obtained for the current time step within $\mathcal{X}_t$, and repeat the above steps.

---

**Algorithm 1** Data generation algorithm

---

1: Initialize a furnace configuration via set points and walk interval.
2: Initialize $\mathcal{X} = \{\}$, $T > 0$ (max no. of steps).
3: Initialize $tG(0), tS\ fur(0), tS\ obs(0)$ with ambient temperatures, and $fr(0)$.
4: **for** t=1 **to** $T$ **do**                                                                                    ▷ t: time step
5:     $fr(t) \leftarrow$ update firing rates$(fr(t-1), \text{set point temperatures}, tG(t-1), tS\ fur(t-1), tS\ obs(t-1))$
6:     $\boldsymbol{q}^{(t)} \leftarrow$ Enthalpy(Flow-pattern$(fr(t))$)
7:     $\overleftarrow{\boldsymbol{GG}}^{(t)}, \overleftarrow{\boldsymbol{GS}}^{(t)}, \overleftarrow{\boldsymbol{SG}}^{(t)}, \overleftarrow{\boldsymbol{SS}}^{(t)} \leftarrow$ DFA$(tG(t-1), tS\ fur(t-1), tS\ obs(t-1), \boldsymbol{GG}, \boldsymbol{GS}, \boldsymbol{SG}, \boldsymbol{SS})$
8:     $tG(t) \leftarrow$ EBV$(\boldsymbol{q}^{(t)}, \overleftarrow{\boldsymbol{GG}}^{(t)}, \overleftarrow{\boldsymbol{GS}}^{(t)})$
9:     $\boldsymbol{w}^{(t)} \leftarrow$ heat-transfer$(tG(t), tS\ fur(t-1), tS\ obs(t-1), \overleftarrow{\boldsymbol{SS}}^{(t)}, \overleftarrow{\boldsymbol{SG}}^{(t)})$
10:     $tS\ fur(t), tS\ obs(t) \leftarrow$ EBS(conduction$(\boldsymbol{w}^{(t)})$), $\boldsymbol{n}^{(t)} \leftarrow$ conduction$(\boldsymbol{w}^{(t)})$
11:     $\mathcal{X}_t \leftarrow \{fr(t), \text{Flow-pattern}(fr(t)), \boldsymbol{q}^{(t)}, tG(t), tS\ fur(t), tS\ obs(t), \boldsymbol{w}^{(t)}, \boldsymbol{n}^{(t)}\}$
12:     $\mathcal{X} \leftarrow \mathcal{X} \cup \mathcal{X}_t$
13: **end for**
14: **return** $\mathcal{X}$

---

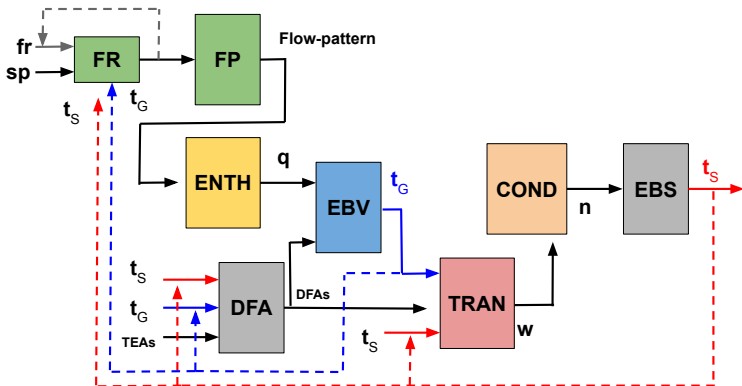

Figure 2: Illustration of flow of the data generation algorithm. The figure is best viewed in color. Dashed lines denote feedback from past time step. Blue/red/gray lines correspond for $t_G$/$t_S$/$fr$, respectively. Block Abbreviations are, FR: Firing Rate, FP: Flow-pattern, ENTH: Enthalpy, TRAN: Heat-transfer, COND: Conduction analysis, EBV/S: Energy-Balance Volume/Surface, and DFA: Directed Flux Area. Details of components present in the main text.

As the temperatures predicted in a time step influence the firing rates for the next time step, there is a time dependency among the data in $\mathcal{X}$. However, most standard off-the-shelf Machine Learning (ML)/ DL models suitable for regression require the data in an Independent and Identically Distributed (IID) format, that could be loaded in a tabular form (with each row being an instance and the columns representing the attributes). Thus, for a particular furnace configuration $c$, to convert $\mathcal{X}$ to IID, we essentially do a vertical stacking of all $\mathcal{X}_t$ within $\mathcal{X}$ to obtain a matrix $\mathbf{X}_c$, whose number of rows equal the number of steps $T > 0$.

Firstly, we do a forward step of all the columns involving the temperature terms and add corresponding copies of the columns to $\mathbf{X}_c$. Then, we do a backward step of the column involving the firing rates, and add the corresponding copy of the column to $\mathbf{X}_c$. We finally remove the first and last rows of the $\mathbf{X}_c$. This helps break the time-dependency among the rows, by introducing previous time-step temperatures, and next firing rates for each of the time steps present in a row. After this, we can vertically stack all the matrices $\mathbf{X}_c$, for all $c$ present in either of train/val/test splits as desired, and perform a random shuffling to obtain a final matrix representing the IID data set $\mathcal{X}_{IID}$. Further details can be found in the Appendix, with respect to implementation details, and number of steps/ IID samples.

We make sure that a furnace configuration $c$ can only belong to either of train/val/test splits. This guarantees that the overall experiments conducted are naturally set for testing the OOD generalizability of a trained model, as the test data is always going to come from a furnace configuration not seen during training. In our framework, different furnace configurations can inherently be considered to be belonging to different data distributions.

### 3.3 (Claim 2/ Regularization) Zone method based PINN

DL models are not naturally good at generalizing to Out-Of-Distribution (OOD) instances (Gulrajani & Lopez-Paz, 2020). In our context, such OOD data could belong to furnace configurations (operating conditions) not seen during training. To tackle this, we propose employing a novel Physics-Informed Neural Network (PINN) model (in our experiments we show its natural extension to both MLP and LSTM architectures). This is done by incorporating prior physical knowledge based on the zone method using a set of our novel proposed Energy-Balance regularizers.

To explain our PINN (see Figure 1b), let us use eq(1) and denote: $\mathcal{L}_{sup} = \mathbb{E}_{(\boldsymbol{x}^{(i)}, \boldsymbol{y}^{(i)}) \in \mathcal{X}_{IID}}[||\boldsymbol{y}^{(i)} - f_\theta(\boldsymbol{x}^{(i)})||_2^2]$ as the standard *supervised term*. Then, the overall PINN loss is formulated as:

$$\mathcal{L}_{total} = \mathcal{L}_{sup} + \lambda_{ebv}\mathcal{L}_{ebv} + \lambda_{ebs}\mathcal{L}_{ebs} \tag{2}$$

Here, $\lambda_{ebv}, \lambda_{ebs} > 0$ are hyper-parameters corresponding to $\mathcal{L}_{ebv}$ and $\mathcal{L}_{ebs}$, such that $\mathcal{L}_{ebv}=||\text{normalize}(\boldsymbol{v}_g)||_2^2$ is our proposed regularizer term corresponding to the **E**nergy-**B**alance equations for the **V**olume zones (**EBV**)

using the zone method. Similarly, $\mathcal{L}_{ebs}=||\text{normalize}(\boldsymbol{v}_s)||_2^2$ is our proposed regularizer term corresponding to the **E**nergy-**B**alance equations for the **S**urface zones (**EBS**). Normalizing an output vector of a neural network in a regression task is standard practice for ensuring convergence. In our work, we use: $\text{normalize}(\boldsymbol{v}) = \boldsymbol{v}/\max(\boldsymbol{v})$, where $\max(\boldsymbol{v})$ is the maximum value from among all components in $\boldsymbol{v}$. We propose to represent $\boldsymbol{v}_g$ and $\boldsymbol{v}_s$ as:

$$\begin{aligned}
\boldsymbol{v}_g &= (\boldsymbol{g}_{(g)arr} + \boldsymbol{s}_{(g)arr} - 4\boldsymbol{g}_{leave} + \boldsymbol{h}_g) \in \mathbb{R}^{|G|} \\
\boldsymbol{v}_s &= (\boldsymbol{s}_{(s)arr} + \boldsymbol{g}_{(s)arr} - \boldsymbol{s}_{leave} + \boldsymbol{h}_s) \in \mathbb{R}^{|S|}
\end{aligned} \tag{3}$$

Here, $|G|/|S|$ denotes the number of Gas/ Surface zones. Intuitively, $\boldsymbol{v}_g$ and $\boldsymbol{v}_s$ are vector representatives corresponding to Energy-Balance equations for gas and surface zones respectively.

Having discussed $\boldsymbol{v}_g$ and $\boldsymbol{v}_s$, we now define the terms used to compute them. Let, $\boldsymbol{g}_{(g)arr} \in \mathbb{R}^{|G|}$ be a vector whose $i^{th}$ entry represents the amount of radiation arriving at the $i^{th}$ gas zone from all the other gas zones, $\boldsymbol{s}_{(g)arr} \in \mathbb{R}^{|G|}$, a vector whose $i^{th}$ entry represents the amount of radiation arriving at the $i^{th}$ gas zone from all the other surface zones, $\boldsymbol{g}_{leave} \in \mathbb{R}^{|G|}$, a vector whose $i^{th}$ entry represents the amount of radiation leaving the $i^{th}$ gas zone, and $\boldsymbol{h}_g \in \mathbb{R}^{|G|}$ a heat term. Also, let $T_{g,j}$ (or $T_g$) and $T_{s,j}$ (or $T_s$) denote the $j^{th}$ gas and surface zone temperatures respectively. Then, following EBV equations, the $i^{th}$ entries of $\boldsymbol{g}_{(g)arr}$, $\boldsymbol{s}_{(g)arr}$, $\boldsymbol{g}_{leave}$ and $\boldsymbol{h}_g$ can be computed as:

$$\begin{aligned}
\boldsymbol{g}_{(g)arr}(i) &= \sum_j^{|G|} \boldsymbol{G_i}\overleftarrow{\boldsymbol{G_j}}\sigma T_{g,j}^4 \\
\boldsymbol{s}_{(g)arr}(i) &= \sum_j^{|S|} \boldsymbol{G_i}\overleftarrow{\boldsymbol{S_j}}\sigma T_{s,j}^4 \\
\boldsymbol{g}_{leave}(i) &= \sum_n^{|N_g|} a_{g,n}(T_{g,i})k_{g,n}\sigma V_i T_{g,i}^4 \\
\boldsymbol{h}_g(i) &= -(\dot{Q}_{conv})_i + (\dot{Q}_{fuel,net})_i + (\dot{Q}_a)_i + \boldsymbol{q}_i
\end{aligned} \tag{4}$$

Here, the constants (known apriori) $(\dot{Q}_{conv})_i$, $(\dot{Q}_{fuel,net})_i$, and $(\dot{Q}_a)_i$ respectively denote the convection heat transfer, heat release due to input fuel, and thermal input from air/ oxygen. An enthalpy vector $\boldsymbol{q} \in \mathbb{R}^{|G|}$ is computed using the flow-pattern obtained via polynomial curve fitting during simulation. $\sigma$ is the Stefan-Boltzmann constant, $V_i$ is volume of $i^{th}$ gas zone.

Let, $\boldsymbol{s}_{(s)arr} \in \mathbb{R}^{|S|}$, be a vector whose $i^{th}$ entry represents the amount of radiation arriving at the $i^{th}$ surface zone from all the other surface zones, $\boldsymbol{g}_{(s)arr} \in \mathbb{R}^{|S|}$, a vector whose $i^{th}$ entry represents the amount of radiation arriving at the $i^{th}$ surface zone from all the other gas zones, $\boldsymbol{s}_{leave} \in \mathbb{R}^{|S|}$, a vector whose $i^{th}$ entry represents the amount of radiation leaving the $i^{th}$ surface zone, and $\boldsymbol{h}_s \in \mathbb{R}^{|S|}$ a heat term. Then, following EBS equations, the $i^{th}$ entries of $\boldsymbol{s}_{(s)arr}$, $\boldsymbol{g}_{(s)arr}$, $\boldsymbol{s}_{leave}$ and $\boldsymbol{h}_s$ can be computed as:

$$\begin{aligned}
\boldsymbol{s}_{(s)arr}(i) &= \sum_j^{|S|} \boldsymbol{S_i}\overleftarrow{\boldsymbol{S_j}}\sigma T_{s,j}^4 \\
\boldsymbol{g}_{(s)arr}(i) &= \sum_j^{|G|} \boldsymbol{S_i}\overleftarrow{\boldsymbol{G_j}}\sigma T_{g,j}^4 \\
\boldsymbol{s}_{leave}(i) &= A_i \epsilon_i \sigma T_{s,i}^4 \\
\boldsymbol{h}_s(i) &= A_i (\dot{q}_{conv})_i - \dot{Q}_{s,i}
\end{aligned} \tag{5}$$

For a surface zone $i$, the constants (known apriori) $A_i(\dot{q}_{conv})_i$ and $\dot{Q}_{s,i}$ respectively denote the heat flux to the surface by convection and heat transfer from it to the other surfaces. Here, $A_i$ is the area, and $\epsilon_i$ is the emissivity of the $i^{th}$ surface zone. In eq(4), since the computations are being done for learning the gas zone related terms, the $T_g$ terms after being obtained from $f_\theta(\boldsymbol{x})$ ($\boldsymbol{x}$: input tensor to the PINN) are kept associated

with the computational graph for back-propagating, but not the $T_s$ terms. The reverse is true in eq(5) where we are learning for the surface zone related terms, i.e., $T_s$ terms are kept in the computational graph for back-propagating, but not $T_g$ terms. In addition, eq(4) and eq(5) also contain the DFAs ($\tilde{\boldsymbol{GS}} \in \mathbb{R}^{|G| \times |S|}$, $\tilde{\boldsymbol{SS}} \in \mathbb{R}^{|S| \times |S|}$, $\tilde{\boldsymbol{GG}} \in \mathbb{R}^{|G| \times |G|}$, and $\tilde{\boldsymbol{SG}} \in \mathbb{R}^{|S| \times |G|}$) and terms such as $a_{g,n}(T_{g,i}), k_{g,n}$, details of which can be referred from the Appendix A.

---

**Algorithm 2** PyTorch-styled pseudo-code for training loop of our PINN

---

```
### TRAINING ###
criterion = nn.MSELoss()
optimizer = optim.Adam(model.parameters(), lr=LEARNING_RATE)
for e in tqdm(range(1, EPOCHS+1)):
    model.train()
    for (batch_idx, sample_batched) in enumerate(train_loader_EBVS):
        #sample_batched[0]:data, sample_batched[1]:labels, sample_batched[2]:auxvars
        X_train_batch = sample_batched[0].to(device)
        y_train_batch = sample_batched[1].to(device)
        auxvars_dict_batch = sample_batched[2]

        dfa_GG_tensor_batch = auxvars_dict_batch['dfa_GG_tensor'].to(device)
        sgarr_plus_hg_tensor_batch = auxvars_dict_batch['sgarr_plus_hg'].to(device)
        dfa_SS_tensor_batch = auxvars_dict_batch['dfa_SS_tensor'].to(device)
        gsarr_plus_hs_tensor_batch = auxvars_dict_batch['gsarr_plus_hs'].to(device)

        optimizer.zero_grad()

        y_train_pred = model(X_train_batch)
        tr_loss_regtmps = criterion(y_train_pred, y_train_batch)

        ## EBV terms
        pb_ebv_pred = get_pb_ebv_pred(
            sgarr_plus_hg_tensor_batch, dfa_GG_tensor_batch,
            y_train_pred[:,:n_gas_zones]
            )
        pb_ebv_actual = torch.zeros(pb_ebv_pred.size()).to(device)

        ## EBS terms
        pb_ebs_pred = get_pb_ebs_pred(
            gsarr_plus_hs_tensor_batch, dfa_SS_tensor_batch,
            y_train_pred[:,n_gas_zones:n_gas_zones+n_fur_surf_zones+n_obs_surf_zones]
            )
        pb_ebs_actual = torch.zeros(pb_ebs_pred.size()).to(device)

        tr_loss_ebv = criterion(pb_ebv_pred, pb_ebv_actual) / y_train_pred.size(0)
        tr_loss_ebs = criterion(pb_ebs_pred, pb_ebs_actual) / y_train_pred.size(0)

        batch_loss=tr_loss_regtmps+lambda_ebv*tr_loss_ebv+lambda_ebs*tr_loss_ebs
        batch_loss.backward()
        optimizer.step()
```

---

In Algorithm 2, we outline the key steps required in training our PINN model. The training involves a typical mini-batch based optimization, where each instance in a mini-batch contains the various entities obtained from one row/time step of the IID data set we discussed earlier. The entities are present in their respective columns. The columns for the constant terms (e.g., $(\dot{Q}_{conv})_i$, $(\dot{Q}_{fuel,net})_i$, $(\dot{Q}_a)_i$, $A_i(\dot{q}_{conv})_i$ and $\dot{Q}_{s,i}$) will have the values repeated across all the corresponding rows.

As observed in Algorithm 2, `X_train_batch` and `y_train_batch` correspond to $\boldsymbol{x}^{(i)}$ and $\boldsymbol{y}^{(i)}$ in $\mathcal{X}_{IID}$, and are used to compute `tr_loss_regtmps` representing $\mathcal{L}_{sup}$ in eq(2). `tr_loss_ebv` and `tr_loss_ebs` respectively correspond to $\mathcal{L}_{ebv}$ and $\mathcal{L}_{ebs}$ in eq(2). The collection of the $T_g$ terms for being associated with the computational graph for back-propagation by virtue of use in eq(4), is done by `y_train_pred[:,:n_gas_zones]`. Similar role towards back-propagation via $T_s$ terms in eq(5) is taken care of by `y_train_pred[:,n_gas_zones:n_gas_zones+n_fur_surf_zones+n_obs_surf_zones]`.

`get_pb_ebv_pred()` computes $\boldsymbol{v}_g$ in eq(3) for each instance (corresponding to a time-step of zone method) present in a mini-batch of the variables obtained from the already created IID data set. In doing so, each of the $|G|$ elements of $\boldsymbol{v}_g$ are computed using eq(4) and the corresponding/relevant auxiliary variables from the IID data. `sgarr_plus_hg_tensor_batch` collects mini-batch terms using relevant terms like $\boldsymbol{s}_{(g)arr}, \boldsymbol{h}_g$

Table 1: Comparison of proposed EBV+EBS method against the naive baseline and MLP without physics-based regularizer, in an IID evaluation setting (on test data). Best result for a row is in bold. ↓/ ↑: lower/ higher metric is better.

| | Without previous temperatures as inputs | | |
| --- | --- | --- | --- |
| | Baseline Methods | | Proposed Physics-Informed Method |
| Performance Metric | Naive Avg | MLP Baseline | EBV+EBS |
| RMSE tG ($\downarrow$) | 58.63 | 10.27 | **10.04** |
| RMSE tS fur ($\downarrow$) | 53.03 | 8.94 | **7.95** |
| RMSE tS obs ($\downarrow$) | 68.19 | **30.94** | 31.64 |
| MAE tG ($\downarrow$) | 39.04 | 7.31 | **7.19** |
| MAE tS fur ($\downarrow$) | 34.45 | 5.97 | **5.58** |
| MAE tS obs ($\downarrow$) | 42.27 | **14.95** | 15.13 |
| $R^2$ tG ($\uparrow$) | -0.031 | 0.954 | **0.961** |
| $R^2$ tS fur ($\uparrow$) | -0.042 | 0.948 | **0.959** |
| $R^2$ tS obs ($\uparrow$) | -0.065 | **0.886** | 0.885 |
| mMAPE fr ($\downarrow$) | 155.30 | 7.41 | **6.84** |
| | With previous temperatures as inputs | | |
| | Baseline Methods | | Proposed Physics-Informed Method |
| Performance Metric | Naive Avg | MLP Baseline | EBV+EBS |
| RMSE tG ($\downarrow$) | 58.63 | 5.75 | **4.91** |
| RMSE tS fur ($\downarrow$) | 53.03 | 4.77 | **4.24** |
| RMSE tS obs ($\downarrow$) | 68.19 | **17.18** | 17.39 |
| MAE tG ($\downarrow$) | 39.04 | 3.21 | **3.01** |
| MAE tS fur ($\downarrow$) | 34.45 | 3.09 | **2.74** |
| MAE tS obs ($\downarrow$) | 42.27 | **4.80** | 5.81 |
| $R^2$ tG ($\uparrow$) | -0.031 | 0.984 | **0.989** |
| $R^2$ tS fur ($\uparrow$) | -0.042 | 0.983 | **0.989** |
| $R^2$ tS obs ($\uparrow$) | -0.065 | **0.966** | 0.966 |
| mMAPE fr ($\downarrow$) | 155.30 | 7.86 | **6.87** |

Table 2: Comparison of proposed EBV+EBS method against MLP without physics-based regularizer, in an auto-regressive evaluation setting (on test data). Best result for a row is in bold. ↓: lower metric is better.

| | Without previous temperatures as inputs | | |
| --- | --- | --- | --- |
| Metric/ Method | MLP | EBV+EBS | EBV+EBS improvement over MLP (in %) |
| RMSE tG ($\downarrow$) | 28.6 | **27.3** | 4.2 |
| RMSE tS fur ($\downarrow$) | 10.1 | **9.6** | 4.8 |
| RMSE tS obs ($\downarrow$) | **42.7** | 44.0 | -3.1 |
| MAE tG ($\downarrow$) | 17.1 | **16.1** | 5.8 |
| MAE tS fur ($\downarrow$) | 7.8 | **7.3** | 6.5 |
| MAE tS obs ($\downarrow$) | **20.0** | 20.2 | -1.1 |
| mMAPE fr ($\downarrow$) | 69.2 | **63.5** | 8.2 |
| | With previous temperatures as inputs | | |
| Metric/ Method | MLP | EBV+EBS | EBV+EBS improvement over MLP (in %) |
| RMSE tG ($\downarrow$) | 74.1 | **36.8** | 50.3 |
| RMSE tS fur ($\downarrow$) | 74.5 | **25.8** | 65.4 |
| RMSE tS obs ($\downarrow$) | 83.3 | **65.3** | 21.5 |
| MAE tG ($\downarrow$) | 48.8 | **29.3** | 39.9 |
| MAE tS fur ($\downarrow$) | 49.7 | **20.8** | 58.2 |
| MAE tS obs ($\downarrow$) | 53.6 | **42.0** | 21.6 |
| mMAPE fr ($\downarrow$) | 96.2 | **40.6** | 57.8 |

in eq(3) towards $\boldsymbol{v}_g$. The relevant DFA terms are collected in tensor `dfa_GG_tensor_batch`. Similarly, we make use of `get_pb_ebs_pred()`, `dfa_SS_tensor_batch`, `gsarr_plus_hs_tensor_batch` for computing $\boldsymbol{v}_s$ in eq(3) and using eq(5). Having obtained the IID dataset like earlier, it only involves sampling mini-batches via appropriate helper functions in any Deep Learning framework (e.g., PyTorch). In Appendix A, we provide few helper functions which can be helpful to further understand the computation of some of the tensors involved in the training loop described.

## 4 Experimental Results

### 4.1 PINN vs MLP vs Naive Baseline

In our experiments, we first study the MLP based architecture. We call our proposed PINN using a MLP architecture as EBV+EBS (due to the energy-balance terms). We compare it against a baseline MLP with the same architecture as our PINN, but without the physics-based EB regularizers (architecture and training details in appendix). We also compare a naive baseline, which, for a test instance, simply predicts the average value of a target variable using the training data. To evaluate the methods, we make use of an IID dataset (details in Appendix). For each test instance, we have input and output ground-truth values. We cast the prediction as a regression problem, and hence we can make use of the following standard regression performance evaluation metrics: Root Mean Squared Error (RMSE), Mean Absolute Error (MAE), and Coefficient of determination ($R^2$). We separately compute a model's performance for gas zone, furnace surface zone and obstacle surface zone temperatures, thus resulting in the metrics RMSE tG, RMSE tS fur, RMSE tS obs, MAE tG, MAE tS fur, MAE tS obs, $R^2$ tG, $R^2$ tS fur, $R^2$ tS obs.

We train all models in the IID training split, tune hyper-parameters using the validation split, and report all performance metrics on the test split. We also predict the next firing rates, and because they are in practice within the normalized range $[0, 1]$ (Hu et al., 2016), we make use of a modified Mean Absolute Percentage Error (mMAPE), by adding a small value $\epsilon = 0.05$ to the denominator of the MAPE computation (to scale up the metric values). A lower value of RMSE, MAE, and mMAPE indicates a better performance (indicated by ↓), while a higher value of $R^2$ indicates a better performance (indicated by ↑). The best obtained metric by a method shall be shown in **bold** in the result tables.

From Table 1, we noticed the superior performance of our proposed PINN EBV+EBS over the MLP, as it better respects the underlying physics. When previous temperatures are provided in the inputs, due to additional signals, performance of both improves, but ours becomes better. In all cases, both the MLP and our PINN performs significantly better than the naive baseline, thereby, highlighting that learning based methods indeed help in this scenario. In addition to DL, we also compare our method against varying classical ML baselines (in an IID evaluation setting), the results of which can be found in the Appendix, in Table 9.

## 4.2 Auto-Regressive (AR) evaluation

We additionally perform evaluation on the test data in an Auto-Regressive (AR) manner: Only for the test data set, rearrange all the test instances of a furnace configuration according to their time step values. Now, use the model checkpoint obtained in an IID manner on the training data, to infer on the test data set for a configuration, and compute performance metrics. The metrics across all configurations in the test split are then averaged.

Specifically, in the inference time step $t$ of the AR evaluation, instead of providing the trained model the input values of firing rates, gas and surface zone temperatures obtained during simulation, we rather use the model predicted values obtained in the inference time step $t-1$. This is similar to a real-world operation, where a deployed model would be expected to continuously predict different values and use them as inputs for the next model predictions.

In the IID evaluation setting (Table 1), at each inference step, the data is sampled IID, and the model inputs are those that are obtained via the simulation, which would be correct, as per the zone model. However, in the AR evaluation, over time, the model inputs being provided by its own predictions done earlier, are prone to cumulative error propagation. Thus, we can see that in Table 2, the values of performance metrics have degraded compared to the metrics obtained in the IID evaluation setting (Table 1). Even then, our PINN EBV+EBS outperforms the baseline MLP, on an average.

Also, when previous temperatures are provided as inputs, this makes the AR evaluation more challenging. This is because there are now more input entities which could be predicted by the model sub-optimally. In this case, performance of the MLP deteriorates significantly. This might be because it merely learns to memorize the training data, without really understanding the underlying physical phenomenon. On the other hand, our method, being aware of the underlying physics, is more generalizable and hence performs significantly better than the MLP baseline (up to 50-65% improvements). In the appendix, we provide additional experimental results, including the in-depth analysis of our PINN EBV+EBS.

## 4.3 Visual analysis of temperature predictions of our PINN

In Tables 1-2, we observed a common trend across various performance metrics (and among the methods compared): the prediction performance is relatively worse for the obstacle surfaces, whereas it is equivalently reasonable for gas zones and furnaces surfaces. This is expected because a single obstacle (e.g., a steel slab) would typically contain 6 surfaces, and the obstacles usually would be placed very close to each other. At the same time, the slabs would be moving towards the discharge end at the right. The close proximity, along with their movement, makes it relatively challenging to accurately predict the temperatures, as compared to the gas zone and furnace surface zone temperatures.

Additionally, a look at Figure 1a also illustrates that the obstacle temperatures should gradually and sharply increase towards the right (as indicated by a darker shade of red). Now, as seen in the top sub-figure of Figure 1a, each of the gaseous regions contribute to two vertical zones. These, along with the surface zones,

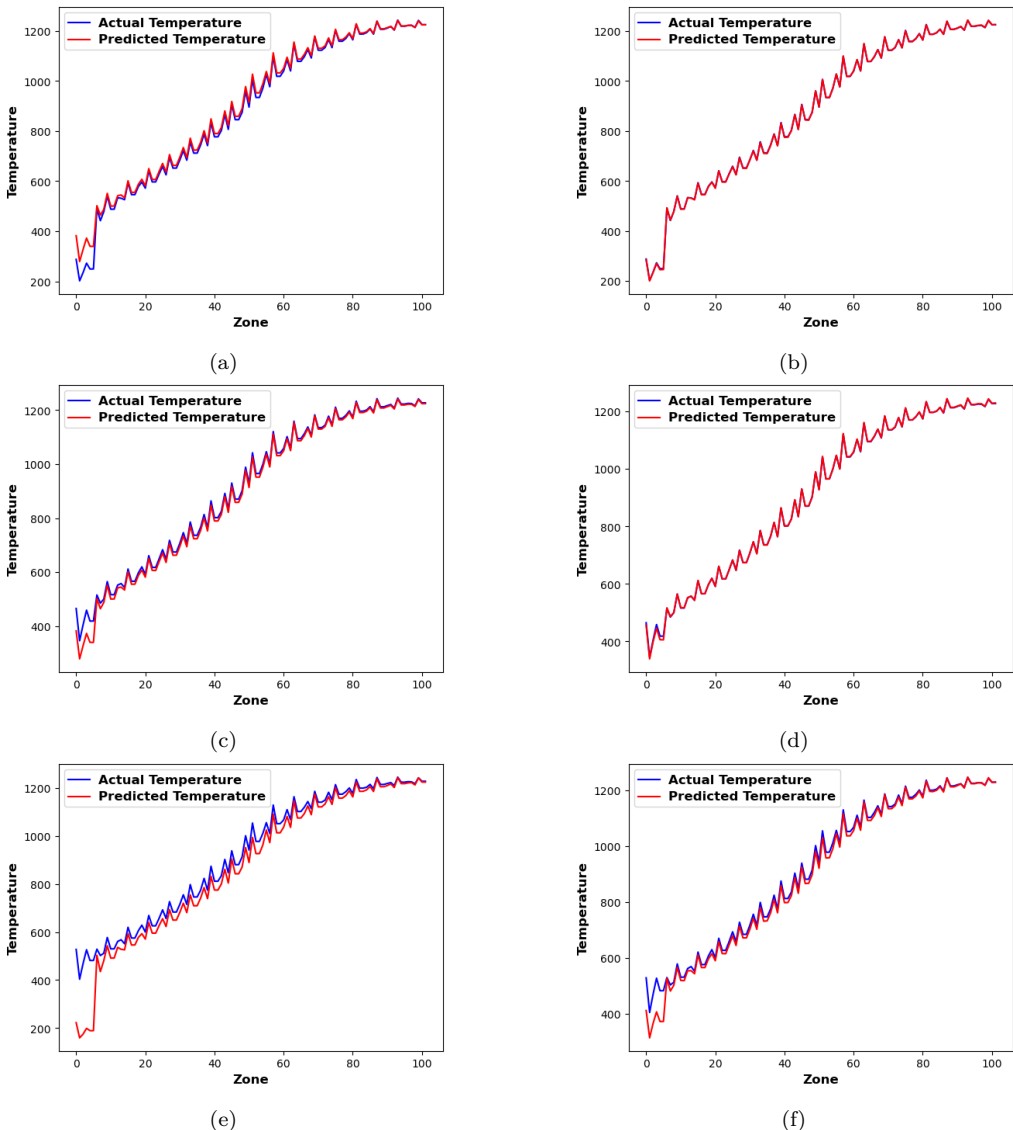

Figure 3: Figure is best viewed in color. Plot of actual (blue) and predicted (red) temperatures (in $^\circ C$) across all obstacle surface zones using our PINN model EBV+EBS on the test data. The left (and resp, right) panels correspond to IID evaluation scenarios without (and resp, with) previous temperatures as inputs. The various rows represent predictions for arbitrary time steps. Predictions in the charging (left) end encounters majority of the misclassifications. Including previous temperatures in the input for the IID scenario is shown to be beneficial, as also observed empirically.

are broadly further distributed among the dark, a few control, and the soaking zones. Across these broad zones, we expect an oscillating pattern for the gas zone and furnace surface zone temperatures, with albeit a minor increasing trend among the gas zones.

Based on the above hypotheses, we try to visually plot the actual (in blue) and predicted (in red) temperatures for our PINN model EBV+EBS in Figure 3 and Figure 6 (the latter is in the Appendix A). In Figure 3, we plot the temperatures for the 102 obstacle surface zones. Here, each of the different rows correspond to different time steps. And within a row, the left plot represents the temperature prediction without the previous temperatures as inputs, and the right plot corresponds to those with the previous temperatures as inputs. All the plots correspond to the IID evaluation setting. As expected, we could see the following: i) there is indeed a sharp increasing trend towards the right, and ii) using the previous temperatures as inputs helps in improving the performance for the IID scenario (which we already empirically established). We noted that in particular, predictions in the charging (left) end encounters majority of the misclassifications.

Similar format is also being followed for the Figure 6 (in the Appendix A). There, we did verify visually the oscillating pattern among the gas zone and furnace surface zone temperatures. Also, the better performance for the case with previous temperatures in the input for the IID setting is also observed there. We also noticed the plots indicating the slight prediction mismatches being more prominent in the gas zones, as opposed to the furnace surface zones. This is due to the fact that the gas zones encounter more energy and mass transfer than the statically arranged furnace surfaces, therby making them relatively more challenging.

We believe that an interesting future direction could be to explicitly focus on improving the predictions for the obstacles by additional constraints, for instance, involving the geometry and other features of the slabs (e.g., material quality). This could be looked at as a future work. One may also try to include further information about the 3D structure of the enclosure within the optimization framework.

### 4.4 Discussion on computational aspects

In general, PINNs and accurate simulators (e.g., CFD models) are two different approaches to solving a physical problem. In terms of computational efficiency, they cannot be compared at the same level. While PINNs could take milliseconds for inference, accurate simulators have difficulty even achieving real-time simulation. Thus, PINNs have the potential to be integrated directly into a control system for real-time control. This is because PINNs are a type of approaches that encode the governing equations of the problem into the network training, whereas, accurate simulators are based on numerical methods that discretize the problem domain and solve the equations on a mesh, which can be time-consuming, and challenging to generate for complex geometries or moving boundaries (such as the furnace studied in our work).

Generally speaking, the zone method is faster and simpler to implement than the CFD method. For example, even with a consumer-level PC, to simulate a 341-min real reheating process, the zone model only takes 5 mins, but CFD models often take several days, if not weeks, to provide *useful* results (Hu et al., 2016). Therefore, in this study, we utilize the zone model to generate training data for PINNs. In future studies, the trained PINNs will be integrated directly into furnace control systems. For our study, typically, generating 1500 timesteps of data for a single furnace using the zone method took about 2 hours, including the time for setting different configurations.

However, talking about the absolute time of a CFD case simulation itself depends on many factors, such as mesh density, sub-model selection, step size settings, and computer hardware configuration. Specific to our case, using the same configuration of PC, CFD simulation of the steady-state operating conditions of each setting takes about 5 hours. So the total time taken is 5 hours multiplied by the number of simulated working conditions. For the simulation of unsteady operating conditions, CFD is currently very difficult to implement, and some simplifications must be made. The specific time consumption depends on the duration of the simulated unsteady process. For the real process of 341 min for the case we studied, CFD would take at least 5 days (vs, 5 min of the zone method). As for the neural-network based implementations, for ML-based inference on a Apple M2 Max 32GB, our PINN takes roughly 0.5s for inferring the entire furnace profile for a single time step instance, given the input variables as discussed.

### 4.5 Comparison with LSTM

We conclude our paper with further experiments on the challenging setting of AR evaluation with the previous temperatures as inputs, where all models including our proposed MLP based PINN EBV+EBS incur the highest prediction error as shown in Table 2. Here, we used the trained models to autoregressively predict the subsequent outputs and use them as inputs for the corresponding next time steps. We now particularly compare our proposed MLP based PINN model EBV+EBS against the Long Short-Term Memory (LSTM) based Recurrent Neural Network (RNN) model, which is inherently designed to model sequential information. Typically, the performance of an LSTM model varies with the sequence length (also called, *look-back window*) considered for inference. In our use-case, owing to the zone method, the outputs of a time step depend on the furnace state of the previous time step. Thus, we use a sequence length of one in the LSTM variants for comparison. This also ensures a fair comparison with the proposed EBV+EBS method which is MLP-based.

Table 3: Comparison of our proposed EBV+EBS method (MLP based PINN) against different LSTM variants (LSTM, LSTM2, DLSTM), in an auto-regressive evaluation setting with previous temperatures as inputs (on test data). For the two top performing LSTM variants (LSTM2, DLSTM), we also report the performance of including our EBV and EBS regularizers jointly with the LSTMs, resulting in two Physics-Based LSTMs (PBLSTM2, PBDLSTM). Best result for a row is in bold. ↓: lower metric is better.

| | | | | | **With previous temperatures as inputs (AR evaluation)** | |
|---|---|---|---|---|---|---|
| | | **LSTM Variants** | | | **Proposed Physics-Informed Variants** | |
| Metric/ Method | LSTM | LSTM2 | DLSTM | EBV+EBS | PBLSTM2 (LSTM2+EBV+EBS) | PBDLSTM (DLSTM+EBV+EBS) |
| RMSE tG ($\downarrow$) | 97.4 | 55.3 | 82.9 | 36.8 | 31.4 | **26.9** |
| RMSE tS fur ($\downarrow$) | 82.9 | 48.5 | 80.1 | 25.8 | 23.4 | **20.2** |
| RMSE tS obs ($\downarrow$) | 99.3 | 77.6 | 90.6 | **65.3** | 69.2 | 65.9 |
| MAE tG ($\downarrow$) | 73.9 | 34.9 | 57.1 | 29.3 | 23.7 | **20.4** |
| MAE tS fur ($\downarrow$) | 63.0 | 30.1 | 54.7 | 20.8 | 18.0 | **15.5** |
| MAE tS obs ($\downarrow$) | 70.1 | 46.8 | 59.3 | 42.0 | 41.5 | **36.5** |
| mMAPE fr ($\downarrow$) | 157.2 | 68.6 | 109.6 | 40.6 | 22.1 | **21.8** |

For the regression task, we can use LSTM layers followed by Fully-Connected (FC) layers. We report the performance comparisons against LSTM, in Table 3. We empirically tried various combinations and found the following three LSTM variants to be suitable baselines for comparison: i) LSTM: a single LSTM layer, followed by a final FC layer, ii) LSTM2: single LSTM layer followed by deeper FC layers, and iii) DLSTM: Deep stacked LSTM. In all LSTM variants, the final FC layer would have sigmoid non-linearity to squeeze the outputs to $[0, 1]$ (as followed throughout the evaluation). The architectural details of all LSTM variants can be found in the Appendix A.

As observed in Table 3, simply stacking deeper LSTM layers (as in DLSTM) does improve the predictive performance over the base LSTM. However, we found that keeping a single LSTM layer, followed by deeper FC layers led to better results among LSTM variants (as seen for LSTM2). We also noted that the LSTM2 variant outperforms the baseline MLP method from Table 2. This is possibly by virtue of its capability to better capture the correlations among the data, due to underlying combination of LSTM and FC layers. However, despite the simpler MLP based architecture, our proposed PINN model EBV+EBS with its physics awareness, still performs better than the top performing LSTM2, based on the average performance. In the Appendix A, we provide additional experiments on LSTM in the IID setting.

### 4.5.1 Physics-based LSTM with our proposed energy-balance regularizers

As described by eq(2), the supervised term $\mathcal{L}_{sup}$ in our proposed framework can employ any generic regression-based model requiring input-output pairs $(\boldsymbol{x}^{(i)}, \boldsymbol{y}^{(i)}) \in \mathcal{X}_{IID}$. Therefore, it can naturally encompass the loss term of a LSTM model. With the auxiliary variables that are present in $\mathcal{X}_{IID}$, we can easily plug the energy-based regularizer terms $\mathcal{L}_{ebv}$ and $\mathcal{L}_{ebv}$ to derive the objective $\mathcal{L}_{total}$ for a physics-based LSTM, to make it aware of the governing physics equations. As we are only adding regularizer terms in our objective, just like the case of MLP to EBV+EBS PINN conversion, we do not require any architectural changes.

To this end, we pick the top two LSTM variants LSTM2 and DLSTM and add our physics-based regularizers in their respective objectives, and name the resulting methods as Physics-Based LSTM2 (PBLSTM2) and Physics-Based DLSTM (PBDLSTM). The base architecture as discussed, remains the same for both. Therefore, any differences in performance can naturally be attributed to the physics-based regularization terms. We do not tune the hyperparameters $\lambda_{ebv}, \lambda_{ebs}$ for PBLSTM2 and PBDLSTM, but simply reuse the ones from EBV+EBS PINN (as described in Appendix A).

In Table 3, we also report the results of the physics-based PBLSTM2 and PBDLSTM methods. We notice that both PBLSTM2 and PBDLSTM obtain a better performance than the corresponding LSTM variants LSTM2 and DLSTM without the physics-based regularizers. This shows the promise of including the physics-awareness during the training. We further observed that both of these also outperform our MLP-based PINN model proposed earlier, i.e., EBV+EBS. This might be attributed to the underlying recurrent LSTM model architecture being able to capture more complex correlation in the data, compared to a MLP architecture.

We noticed an interesting observation: while the LSTM2 model inherently was performing better than the DLSTM model, it is the DLSTM that gains a better performance than the LSTM2 model when plugged with physics-based regularizers, as evident by the lower errors obtained for PBDLSTM over PBLSTM2. We believe that for the same data set, increasing the model complexity by introducing more LSTM layers in DLSTM was leading to overfitting, and as a result, a single LSTM layer with deeper FC layers instead was more beneficial in LSTM2. However, with further physics based regularizers, the training with deeper LSTM layers in DLSTM proved beneficial in learning complex relations, while better adhering to the governing laws of physics by virtue of its energy-balance based regularizers.

## 5  Conclusion

In this paper, we visit the classical Hottel's zone method from a Machine Learning (ML) perspective. Our first contribution involved aspects related to data generation for ML/DL model training via regression to tackle the temperature prediction within high energy processes such as reheating furnaces. We specifically present an algorithm to generate temporal data via the zone method, and then discuss how the same could be recasted as an IID data set. With the data in place, we empirically evaluate a range of ML/DL techniques. For DL based methods, we consider both MLP as well as LSTM architectures. However, to better generalize to OOD data (in this case, data coming from a different furnace configuration, as opposed to that seen in training), we further propose a physics-based regularization inspired by the Energy-Balance equations involved in the zone method. This results in a PINN model, which is shown to be empirically better than the baselines compared against, across a wide range of regression based performance metrics. We also visually inspect the predicted temperature profiles for various zones, and found them to be adhering to our hypotheses based on the furnace geometry studied. In the future, our work could be extended for newer avenues, such as incorporating additional furnace geometries via transfer learning and continual learning, or adding obstacle-specific additional constraints in the optimization framework, to name a few.

**Broader Impact Statement**

Not applicable.

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

# A  Appendix

**Exchange Factors, TEA, DFA, and WSGG:**  The first step in the Zone method involves computation of Exchange Factors (Yuen & Takara, 1997). The exchange factor among a pair of volume zones $V_i$ and $V_j$ is expressed as:

$$g_i g_j = \int_{V_i} \int_{V_j} \frac{k_i k_j e^{-\tau} dV_i dV_j}{\pi r^2} \tag{6}$$

Physically, it represents the energy radiated from $V_i$ and absorbed/ scattered by $V_j$. Here, $k$ denotes the respective extinction coefficient, $\tau$ is the optical thickness among differential volume elements $dV_i$ and $dV_j$, and $r = \sqrt{(x_i - x_j)^2 + (y_i - y_j)^2 + (z_i - z_j)^2}$. Now, let $\boldsymbol{n_i}$ and $\boldsymbol{n_j}$ respectively be unit normal vectors of $dA_i$ and $dA_j$ (corresponding to two surface zones $A_i$ and $A_j$). Then, the exchange factors $g_i s_j$ (between volume zone $V_i$ and surface zone $A_j$) and $s_i s_j$ (between surface zone $A_i$ and surface zone $A_j$), can be expressed as:

$$g_i s_j = \int_{V_i} \int_{A_j} \frac{k_i |\boldsymbol{n_j}.r| e^{-\tau} dV_i dA_j}{\pi r^3} \tag{7}$$

$$s_i s_j = \int_{A_i} \int_{A_j} \frac{|\boldsymbol{n_i}.r||\boldsymbol{n_j}.r| e^{-\tau} dA_i dA_j}{\pi r^4} \tag{8}$$

Numerical evaluation of the above equations being complex, has led to analytical approximations, by considering an enclosure as a cube-square system, i.e, by representing a volume as a cube, and a surface as a square. This facilitates the tabulation of a "generic" set of exchange factors, which are applicable for most practical industrial geometries, using an updated Monte-Carlo based Ray-Tracing (MCRT) algorithm

(Matthew et al., 2014). To this end, such pre-computed generic values are refered to as Total Exchange Areas (TEA), and we denote them by: $\overline{G_iS_j}$, $\overline{S_iS_j}$, $\overline{G_iG_j}$ and $\overline{S_iG_j}$. Here, $\overline{S_iG_j} = \overline{G_iS_j}$. Note that throughout the text, G(or g) and S(or s) shall indicate terms corresponding to Gas/Volume, and Surface respectively.

To account for our formulation of a neural network based approach, we first introduce the following four tensors to collectively represent the above TEAs: $\boldsymbol{GS} \in \mathbb{R}^{|G| \times |S| \times |N_g|}$, $\boldsymbol{SS} \in \mathbb{R}^{|S| \times |S| \times |N_g|}$, $\boldsymbol{GG} \in \mathbb{R}^{|G| \times |G| \times |N_g|}$, $\boldsymbol{SG} \in \mathbb{R}^{|S| \times |G| \times |N_g|}$. Here, $|G|$, $|S|$ respectively denote the number of gas/ volume zones, and number of surface zones. In practice, $|N_g|$ gases representing real gas medium are used, and hence, a third dimension has also been used in the above tensors. As discussed above, TEAs are pre-computed constants, used as inputs to our model. Slightly abusing notations, we can refer to a TEA by considering only the first two dimensions (for a pair of zones).

The next step is to compute the Radiation Exchange factors, or the Directed Flux Areas (DFA), considering radiating gas medium through a Weighted Sum of the mixed Grey Gases (WSGG) model (Hu et al., 2016):

$$G_i\overleftarrow{G_j} = \sum_{n=1}^{N_g} a_{g,n}(T_{g,j})(\overline{G_iG_j})_{k=k_n} \tag{9}$$

$$S_i\overleftarrow{S_j} = \sum_{n=1}^{N_g} a_{s,n}(T_{s,j})(\overline{S_iS_j})_{k=k_n} \tag{10}$$

$$G_i\overleftarrow{S_j} = \sum_{n=1}^{N_g} a_{s,n}(T_{s,j})(\overline{G_iS_j})_{k=k_n} \tag{11}$$

$$S_i\overleftarrow{G_j} = \sum_{n=1}^{N_g} a_{g,n}(T_{g,j})(\overline{S_iG_j})_{k=k_n} \tag{12}$$

Here, $\leftarrow$ indicates the direction of flow. $T_{g,j}$ and $T_{s,j}$ denote the temperatures for the $j^{th}$ volume and surface zones respectively, and are the values we want our model to predict. Note that the collective representation of the DFAs can be expressed as: $\tilde{\boldsymbol{GS}} \in \mathbb{R}^{|G| \times |S|}$, $\tilde{\boldsymbol{SS}} \in \mathbb{R}^{|S| \times |S|}$, $\tilde{\boldsymbol{GG}} \in \mathbb{R}^{|G| \times |G|}$, $\tilde{\boldsymbol{SG}} \in \mathbb{R}^{|S| \times |G|}$. In Eq (9)-(12), the TEA terms correspond to a particular grey gas being used, for example, $(\overline{G_iG_j})_{k=k_n}$ represents the TEA $\overline{G_iG_j}$ with the $n^{th}$ gas.

WSGG is a method used to represent the absorptivity/ emissivity of real combustion products with a mixture of a couple of grey gases plus a clear gas, i.e, the number of grey gases is equal to $N_g - 1$. The weighting coefficient $a_{g,n}(T_g)$ is usually expressed as a $N_g^{th}$ order polynomial in $T_g$:

$$a_{g,n}(T_{g,j}) = \sum_{i=0}^{N_g} b_{i+1,n} T_{g,j}^i \tag{13}$$

For each gas indexed by $n$, we have a set of pre-computed correlation coefficients $\{b_{i+1,n}\}_{i=0}^{N_g}$, and an absorption coefficient $k_{g,n}$. Similarly, we can also define the counterpart for surfaces as (with the same correlation coefficients as above for gases):

$$a_{s,n}(T_{s,j}) = \sum_{i=0}^{N_g} b_{i+1,n} T_{s,j}^i \tag{14}$$

We propose the following compact matrix form to collectively represent all DFAs of Eq (11):

$$\tilde{\boldsymbol{GS}} = \sum_{n=1}^{N_g} (\overline{GS})_n \odot \text{broadcast}(\boldsymbol{a}_n^\top) \tag{15}$$

$$\boldsymbol{a}_n = \tilde{b}_n(\boldsymbol{t}_S) \tag{16}$$

Here, $(\overline{GS})_n$ is the $n^{th}$ slice of $\boldsymbol{GS}$ along the third dimension. broadcast$(\boldsymbol{a}_n^\top)$ reshapes $\boldsymbol{a}_n^\top$ to the same dimension as $(\overline{GS})_n$, i.e., $\mathbb{R}^{|G| \times |S|}$. $\boldsymbol{t}_S \in \mathbb{R}^{|S|}$ is a vector containing all the surface zone temperatures, such that its $j^{th}$ entry $\boldsymbol{t}_S(j) = T_{s,j}$. The $j^{th}$ entry $\boldsymbol{a}_n(j)$ of $\boldsymbol{a}_n \in \mathbb{R}^{|S|}$ is computed using the function $\tilde{b}_n$ with the correlation coefficients $\{b_{i+1,n}\}_{i=0}^{N_g}$ as the parameters, and by following eq (14). Similarly, Eq (9), (10) and (12) could also be converted to matrix forms as shown below.

The compact matrix form to collectively represent all DFAs of Eq (9) is expressed as:

$$\overset{\leftarrow}{\boldsymbol{G}}\boldsymbol{G} = \sum_{n=1}^{N_g} (\overline{GG})_n \odot \text{broadcast}(\tilde{b}_n(\boldsymbol{t}_G)^\top) \tag{17}$$

The compact matrix form to collectively represent all DFAs of Eq (10) is expressed as:

$$\overset{\leftarrow}{\boldsymbol{S}}\boldsymbol{S} = \sum_{n=1}^{N_g} (\overline{SS})_n \odot \text{broadcast}(\tilde{b}_n(\boldsymbol{t}_S)^\top) \tag{18}$$

The compact matrix form to collectively represent all DFAs of Eq (12) is expressed as:

$$\overset{\leftarrow}{\boldsymbol{S}}\boldsymbol{G} = \sum_{n=1}^{N_g} (\overline{SG})_n \odot \text{broadcast}(\tilde{b}_n(\boldsymbol{t}_G)^\top) \tag{19}$$

Table 4: Dataset details. A total of 50 configurations have been used which are categorized as normal or abnormal.

| Normal Behaviour Configurations (SP1<SP2<SP3) | | | |
|---|---|---|---|
| **Type 1 (Varying SP1 only)** | **Type 2 (Varying SP2 only)** | **Type 3 (Varying SP3 only)** | **Type 4 (Varying WI only)** |
| 905_1220_1250_750.csv (Training) 915_1220_1250_750.csv (Val) 925_1220_1250_750.csv 935_1220_1250_750.csv (Training) 945_1220_1250_750.csv (Val) 965_1220_1250_750.csv 975_1220_1250_750.csv (Training) 985_1220_1250_750.csv (Val) 995_1220_1250_750.csv | 955_1170_1250_750.csv (Training) 955_1180_1250_750.csv (Val) 955_1190_1250_750.csv 955_1200_1250_750.csv (Training) 955_1210_1250_750.csv (Val) 955_1230_1250_750.csv 955_1240_1250_750.csv (Training) | 955_1220_1230_750.csv (Training) 955_1220_1240_750.csv (Val) 955_1220_1250_750.csv 955_1220_1260_750.csv (Training) 955_1220_1270_750.csv (Val) 955_1220_1280_750.csv 955_1220_1290_750.csv (Training) 955_1220_1300_750.csv | 955_1220_1250_675.csv (Training) 955_1220_1250_690.csv (Val) 955_1220_1250_705.csv 955_1220_1250_720.csv (Training) 955_1220_1250_735.csv (Val) 955_1220_1250_765.csv 955_1220_1250_780.csv (Training) 955_1220_1250_795.csv (Val) 955_1220_1250_810.csv 955_1220_1250_825.csv (Training) |
| Abnormal Behaviour Configurations/ Arbitrary SPs | | | | |
| **Type 1 (start@955-incr-dec/const)** | **Type 2 (start@1220-incr-dec)** | **Type 3 (start@1220-dec-inc)** | **Type 4 (start@1250-dec-inc)** | **Type 5 (start@1250-dec-inc)** |
| 955_1220_1200_750.csv (Training) 955_1220_1210_750.csv (Val) 955_1220_1220_750.csv 955_1250_1220_750.csv (Training) 955_1250_1220_765.csv (Val) 955_1250_1250_750.csv 955_1260_1250_750.csv (Training) 955_1270_1250_750.csv | 1220_1250_955_750.csv (Training) 1220_1250_955_795.csv | 1220_955_1250_750.csv (Training) 1220_955_1250_780.csv | 1250_955_1220_750.csv (Training) 1250_955_1220_825.csv | 1250_1220_955_750.csv (Training) 1250_1220_955_810.csv |

**Data set:** For the IID data set generation, we make use of a FORTRAN code provided by Hu et al. (2016), to represent various furnace configurations of the real-world furnace shown in Figure 1a. We consider 50 different configurations, and create disjoint train-val-test splits in such a way that there is no overlap in the data across different splits. Also, each configuration could belong to either of train/val/test split. As val/test data belong to furnace configurations different from that of training, it naturally makes the test data OOD in nature. Each configuration can be defined by set point temperatures and the walk-interval. Set point temperatures are essentially the desired temperatures that the furnace is expected to achieve at different stages/ zones.

We represent a configuration as: `SP1_SP2_SP3_WI`, where `SP1`, `SP2`, `SP3` and `WI` respectively denote the set point 1, set point 2, set point 3, and walk interval. Note that we consider configurations with both normal conditions (`SP1<SP2<SP3`, as naturally occurring in practice), as well as abnormal ones (arbitrary set points). The details are present in Table 4. Here, each configuration is represented by a .csv file containing 1500 time steps (and with the appropriate training/val label in parenthesis, and no label for a test split). Within a configuration, each time step is sampled with a 15s delay, to account for conduction analysis.

As discussed, to convert $\mathcal{X}$ to $\mathcal{X}_{IID}$, we essentially add new columns and shift the entries. Particularly, we add a new column `firing_rates_next` by shifting the original firing rates column a step back and then dropping the last row. Likewise, we add new columns for *prev* temperatures by shifting the original temperature columns a step forward and then dropping the first row.

The full list of entities that we generate for a time step is: `'timestep'`, `'firing_rates'`, `'walk_interval'`, `'setpoints'`, `'flowpattern'`, `'q_enthalpy'`, `'tG_gaszone'`, `'tS_furnace'`, `'tS_obstacle'`, `'w_flux_furnace'`, `'w_flux_obstacle'`, `'nodetmp_1d_furnace'`, `'nodetmp_2d_obstacle'`. The names of the entities are self-explanatory (e.g., `'nodetmp_1d_furnace'` refers to 1D node temperatures for furnace surfaces, `'nodetmp_2d_obstacle'` refers to 2D node temperatures for obstacle surfaces), where G as usual, denotes *gas zone* and S denotes *surface zone*, the latter, is further divided into *furnace* and *obstacle*. As the temperatures predicted in a time step/ row influence the firing rates for the next time step, there is a time dependency among the rows. However, most standard off-the-shelf ML/ DL models suitable for regression require the data in an Independent and Identically Distributed (IID) format.

---

**Algorithm 3** PyTorch-styled pseudo-code for helper functions in our PINN implementation

---

```
1
2    ### HELPER FUNCTIONS ###
3
4    # For EBV
5    dfa_GG_tensor_all = get_dfa_AB_tensor_all(
6        tea_GG, get_torch_float(X_tG_gaszone_prev).to(device)
7        )
8    sgarr_plus_hg_all = get_sgarr_plus_hg_all(
9        get_torch_float(X_hg).to(device), tea_GS,
10       torch.hstack(( get_torch_float(X_tS_furnace_prev),
11           get_torch_float(X_tS_obstacle_prev) )).to(device)
12           )
13
14   def get_pb_ebv_pred_instance(sgarr_plus_hg_tensor, dfa_GG_tensor, tG_single_pred):
15       ## computes \mathbf{v}_g vector for one time step
16
17   def get_pb_ebv_pred(sgarr_plus_hg_tensor_batch, dfa_GG_tensor_batch, y_train_pred_only_tG):
18       ## calls get_pb_ebv_pred_instance for all instances in the batch
19
20   # For EBS
21   dfa_SS_tensor_all = get_dfa_AB_tensor_all(
22       tea_SS, get_torch_float(np.hstack(
23           [X_tS_furnace_prev, X_tS_obstacle_prev]
24           )).to(device))
25   gsarr_plus_hs_all = get_gsarr_plus_hs_all(
26       get_torch_float(X_hs).to(device), tea_SG,
27       get_torch_float(X_tG_gaszone_prev).to(device)
28       )
29
30   def get_pb_ebs_pred_instance(gsarr_plus_hs_tensor, dfa_SS_tensor, tS_single_pred):
31       ## computes \mathbf{v}_s vector for one time step
32
33   def get_pb_ebs_pred(gsarr_plus_hs_tensor_batch, dfa_SS_tensor_batch, y_train_pred_only_tS):
34       ## calls get_pb_ebs_pred_instance for all instances in the batch
```

---

Assuming that the original time-dependent data is stored in a Pandas DataFrame (using a Python syntax), for each time step we also need the following entities, to obtain IID data: `'firing_rates_next'`, `'tG_gaszone_prev'`, `'tS_furnace_prev'`, and `'tS_obstacle_prev'`. This is because, for computing the entities in a time step, we make use of the temperatures in the previous time step. At the same time, for experimental purposes, we also try to directly predict the next firing rate via ML. Thus, using Python syntax, we could perform the following to generate the IID data:
a) `df['firing_rates_next'] = df['firing_rates'].shift(-1)`
followed by `df = df.drop(df.tail(1).index)`.
b) `df['tG_gaszone_prev']=df['tG_gaszone'].shift(1)`,
`df['tS_furnace_prev'] = df['tS_furnace'].shift(1)`,
`df['tS_obstacle_prev'] = df['tS_obstacle'].shift(1)`
followed by `df = df.drop(df.head(1).index)`.

Essentially, we add a new column `'firing_rates_next'` by shifting the original firing rates column a step back and then dropping the last row. Likewise, we add new columns for *prev* temperatures by shifting the original temperature columns a step forward and then dropping the first row. Please note that some additional auxiliary variables are used by the computational method of Hu et al. (2016), which are mostly constants, and could thus be repeated/ copied for each time step. They are: `'corrcoeff_b'`, `'Qconvi'`, `'extinctioncoeff_k'`, `'gasvolumes_Vi'`, `'QfuelQa_sum'`, `'surfareas_Ai'`, `'emissivity_epsi'`, `'convection_flux_qconvi'`. We later leverage them in training our PINN, with the help of regularizers.

After rearranging the data as IID, we consolidate all the 20 training, 12 validation, and 18 test configurations (with 1500 minus 2 time steps per configuration), resulting in 29960 train, 17976 val, and 26964 test time steps/ IID samples. The 2 time steps are subtracted to account for the shift operations discussed during the IID data creation.

---

**Algorithm 4** PyTorch-styled pseudo-code for additional helper functions in our PINN implementation

---

```
1
2  ### HELPER FUNCTIONS (set 2) ###
3
4  def inverse_transform_Vectorized_pt(scaledtensor,range,min_along_dims,dist):
5      range_min,range_max=range
6      origtensor = min_along_dims+dist*(scaledtensor-range_min)/(range_max - range_min)
7      return origtensor
8
9  def get_an_mat_tensor(tB_singlerow_tensor):
10     tMat_tensor=torch.tile(tB_singlerow_tensor, (Ng, 1))
11     coef_b_mat_T=coef_b_mat.T
12     for ii in range(coef_b_mat_T.shape[1]):# Taylor series loop
13         bn=coef_b_mat_T[:,[ii]]
14         bn_tensor=torch.from_numpy(bn).float().to(device)
15         if ii==0:
16             an_mat_tensor=torch.mul(torch.tile(
17                 bn_tensor, (1, tMat_tensor.size(1))),tMat_tensor**ii)
18         else:
19             an_mat_tensor+=torch.mul(torch.tile(
20                 bn_tensor, (1, tMat_tensor.size(1))),tMat_tensor**ii)
21     return an_mat_tensor
22
23  def get_pb_ebv_pred_instance(sgarr_plus_hg_tensor,dfa_GG_tensor,tG_single_pred):
24      startid_col,endid_col=0,n_gas_zones
25
26      tG_current_tensor = inverse_transform_Vectorized_pt(
27          tG_single_pred,(0,1),ytr_min_along_dims[[0],startid_col:endid_col].to(device),
28          ytr_dist[[0],startid_col:endid_col].to(device))
29
30      ggarr_tensor=torch.sum(torch.mul( dfa_GG_tensor , sbcons*torch.tile(
31          tG_current_tensor**4, (dfa_GG_tensor.size(0), 1)) ),1, keepdim=True).T
32
33      an_mat_G_tensor=get_an_mat_tensor(tG_current_tensor)
34
35      tmpmat2=sbcons*torch.mul( torch.tile(
36          Vi_current_tensor ,(an_mat_G_tensor.size(0),1) ) ,
37          torch.tile(tG_current_tensor**4, (an_mat_G_tensor.size(0), 1)) )
38      tmpmat1=torch.mul( an_mat_G_tensor , torch.tile(
39          coef_k_mat_T_tensor, (1,an_mat_G_tensor.size(1))) )
40      gleave_tensor=torch.sum(torch.mul(tmpmat1,tmpmat2),0,keepdim=True)
41
42      pb_ebv_pred_instance= torch.abs(ggarr_tensor+sgarr_plus_hg_tensor-4*gleave_tensor)
43      pb_ebv_pred_instance/=pb_ebv_pred_instance.max(dim=1, keepdim=True)[0]
44
45
46      return pb_ebv_pred_instance
```

---

**Pseudo-codes of helper functions:** In Algorithms 3-4, we provide a few helper functions which can be helpful to further understand the computation of some of the tensors involved in the training loop described in Algorithm 2.

**Training details and model architecture of MLP based PINN EBV+EBS:** We train our MLP based PINN model EBV+EBS for 10 epochs using PyTorch, with early stopping to avoid over-fitting. For the EB equations, we perform the same normalization for enthalpy, flux, and temperatures, as in the final

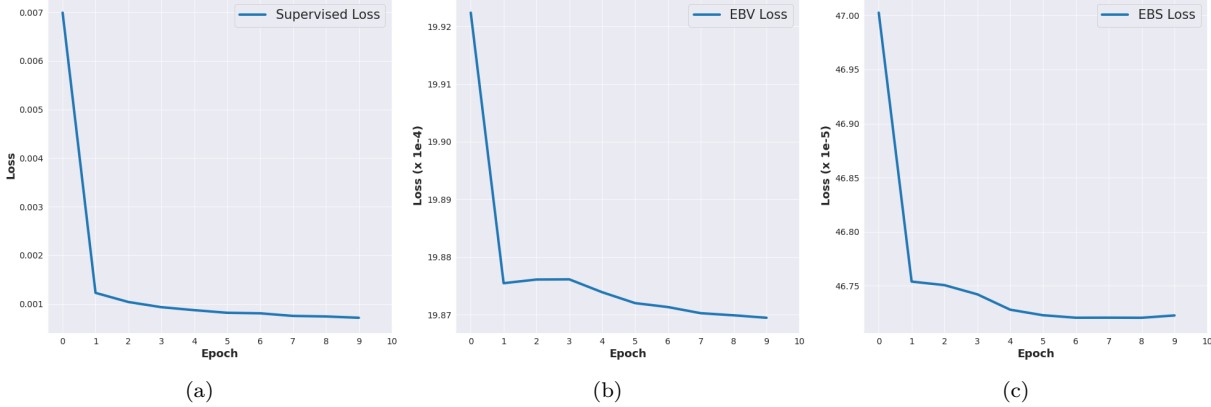

|   (a)   |   (b)   |   (c)   |

Figure 4: Convergence of our EBV+EBS in training, considering: a) Supervised, b) EBV, and c) EBS terms.

Table 5: Performance of EBV+EBS (ReLU) variant of our method against varying hidden layer configurations (IID setting, on test data).

| Metric/ Hidden layer configuration | [100] | [50,100] | **[50,100, 200]** | [50,100, 200,200] | [50,100, 200,200, 205,205] |
|---|---|---|---|---|---|
| RMSE tG ($\downarrow$) | 11.64 | 17.25 | **10.04** | 10.84 | 14.27 |
| RMSE tS fur ($\downarrow$) | 10.05 | 15.23 | 7.95 | **7.83** | 12.46 |
| RMSE tS obs ($\downarrow$) | 34.82 | 37.62 | **31.64** | 33.57 | 36.42 |
| mMAPE fr ($\downarrow$) | 8.76 | 9.15 | **6.84** | 8.06 | 7.51 |

Table 6: Performance of the proposed EBV+EBS variant using different batch sizes (IID setting, on test data).

| Metric | EBV+EBS ReLU bsz=32 | EBV+EBS ReLU bsz=64 | EBV+EBS ReLU bsz=128 |
|---|---|---|---|
| RMSE tG ($\downarrow$) | 12.70 | **10.04** | 10.73 |
| RMSE tS fur ($\downarrow$) | 9.14 | **7.95** | 9.69 |
| RMSE tS obs ($\downarrow$) | 39.75 | **31.64** | 31.79 |
| mMAPE fr ($\downarrow$) | **5.24** | 6.84 | 8.29 |

Table 7: Effect of individual regularizer terms in EBV+EBS (IID setting, on test data).

| Metric | EBV only | EBS only | EBV+EBS |
|---|---|---|---|
| RMSE tG ($\downarrow$) | 11.85 | 11.66 | **10.04** |
| RMSE tS fur ($\downarrow$) | 10.36 | 11.07 | **7.95** |
| RMSE tS obs ($\downarrow$) | 32.46 | 32.04 | **31.64** |
| mMAPE fr ($\downarrow$) | **6.42** | 7.53 | 6.84 |

Table 8: Performance of EBV+EBS using different activation functions in the underlying network (IID setting, on test data).

| Metric | EBV+EBS ReLU | EBV+EBS GeLU | EBV+EBS SiLU | EBV+EBS Hardswish | EBV+EBS Mish |
|---|---|---|---|---|---|
| RMSE tG ($\downarrow$) | **10.04** | 13.57 | 10.07 | 15.26 | 10.16 |
| RMSE tS fur ($\downarrow$) | 7.95 | 8.86 | 8.02 | 14.02 | **7.71** |
| RMSE tS obs ($\downarrow$) | 31.64 | 39.65 | 31.64 | 36.23 | **31.63** |
| mMAPE fr ($\downarrow$) | 6.84 | **5.88** | 6.23 | 7.03 | 6.33 |

neural network output as discussed earlier. We found a learning rate of 0.001 with Adam optimizer and batch size of 64 to be optimal, along with ReLU non-linearity.

We pick the [50,100,200] configuration for hidden layers, i.e., 3 hidden layers, with 50, 100, and 200 neurons respectively. We use $\lambda_{ebv} = \lambda_{ebs} = 0.1$. In general, a value lesser than 1 is observed to be better, otherwise, the model focuses less on the regression task. Following are values of other variables: $|G| = 24$, $|S| = 178$ (76 furnace surface zones and 102 obstacle surface zones), $N_g = 6$, and Stefan-Boltzmann constant=5.6687e-08. Unless otherwise stated, this is the setting we use to report any results for our method, for example, while comparing with other methods.

**In-depth analysis of MLP based PINN EBV+EBS:** To study our PINN in detail, we vary different aspects of our method (e.g., the impact of individual loss/regularization terms, hidden layer configuration, batch size, and activation functions). At a time, we vary one focused aspect, and fix all other hyper-parameters as per the default setting prescribed above.

Firstly, we study the empirical convergence of the default setting of our EBV+EBS method. Fig 4 plots the convergence behaviour of each of the loss terms individually (supervised, EBV, and EBS). Our method, as shown, enjoys a good convergence. In Table 5, we report the performance of our method by varying the hidden layer configurations (e.g., [100] denotes one hidden layer with 100 neurons, [50, 100] denotes two hidden layers with 50, and 100 neurons respectively, and so forth). The maximum values for each row (corresponding to a metric) are shown in bold. We found that it suffices to use [50, 100, 200] configuration for a competitive

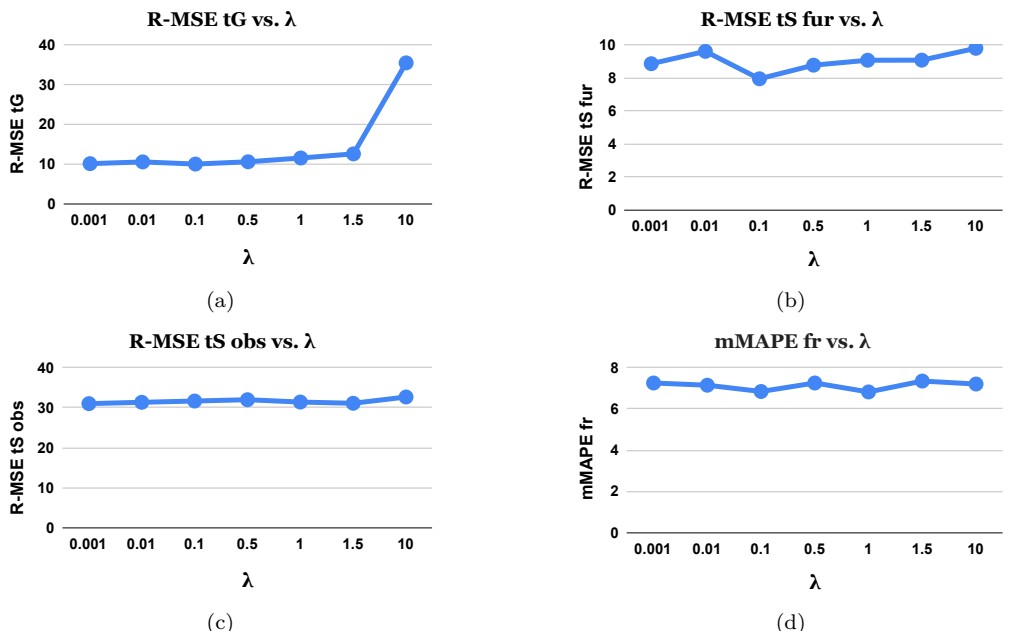

Figure 5: Performance metrics against varying $\lambda$ in MLP based PINN EBV+EBS, where $\lambda_{ebv} = \lambda_{ebs} = \lambda$.

Table 9: Comparison of our proposed method against classical ML baselines, in an IID evaluation setting (on test data). Best result for a row is in bold. ↓: lower metric is better. ↑: higher metric is better.

| | | Classical ML Baselines | | Proposed Physics-Informed Method |
|---|---|---|---|---|
| **Without previous temperatures as inputs** | | | | |
| Performance Metric | DT | RF | H-GBoost | EBV+EBS |
| RMSE tG (↓) | 12.84 | 12.24 | 14.06 | **10.04** |
| RMSE tS fur (↓) | 9.42 | 8.97 | 10.09 | **7.95** |
| RMSE tS obs (↓) | 42.86 | 42.06 | 42.73 | **31.64** |
| $R^2$ tG (↑) | 0.943 | 0.948 | 0.925 | **0.961** |
| $R^2$ tS fur (↑) | 0.951 | 0.957 | 0.934 | **0.959** |
| $R^2$ tS obs (↑) | 0.788 | 0.798 | 0.763 | **0.885** |
| mMAPE fr (↓) | 5.50 | 5.30 | **2.32** | 6.84 |
| **With previous temperatures as inputs** | | | | |
| Performance Metric | DT | RF | H-GBoost | EBV+EBS |
| RMSE tG (↓) | 11.17 | 6.96 | 5.00 | **4.91** |
| RMSE tS fur (↓) | 10.24 | 6.15 | 6.12 | **4.24** |
| RMSE tS obs (↓) | 43.05 | 32.81 | 23.01 | **17.39** |
| $R^2$ tG (↑) | 0.925 | 0.979 | **0.989** | **0.989** |
| $R^2$ tS fur (↑) | 0.915 | 0.977 | 0.983 | **0.989** |
| $R^2$ tS obs (↑) | 0.729 | 0.890 | 0.937 | **0.966** |
| mMAPE fr (↓) | 6.98 | 8.09 | **0.76** | 6.87 |

performance. In Table 6, we vary the batch size in our method. We found a batch size of 64 to provide an optimal performance for our experiments.

In Figure 5, we provide a detailed sensitivity analysis of the hyperparameter values $\lambda_{ebv}$ and $\lambda_{ebs}$ involved in the regularization terms. We set them to an equal value of $\lambda$ and vary the same, to study the effect on the various performance metrics: R-MSE tG, R-MSE tS fur, R-MSE tS obs and mMAPE fr. Though the

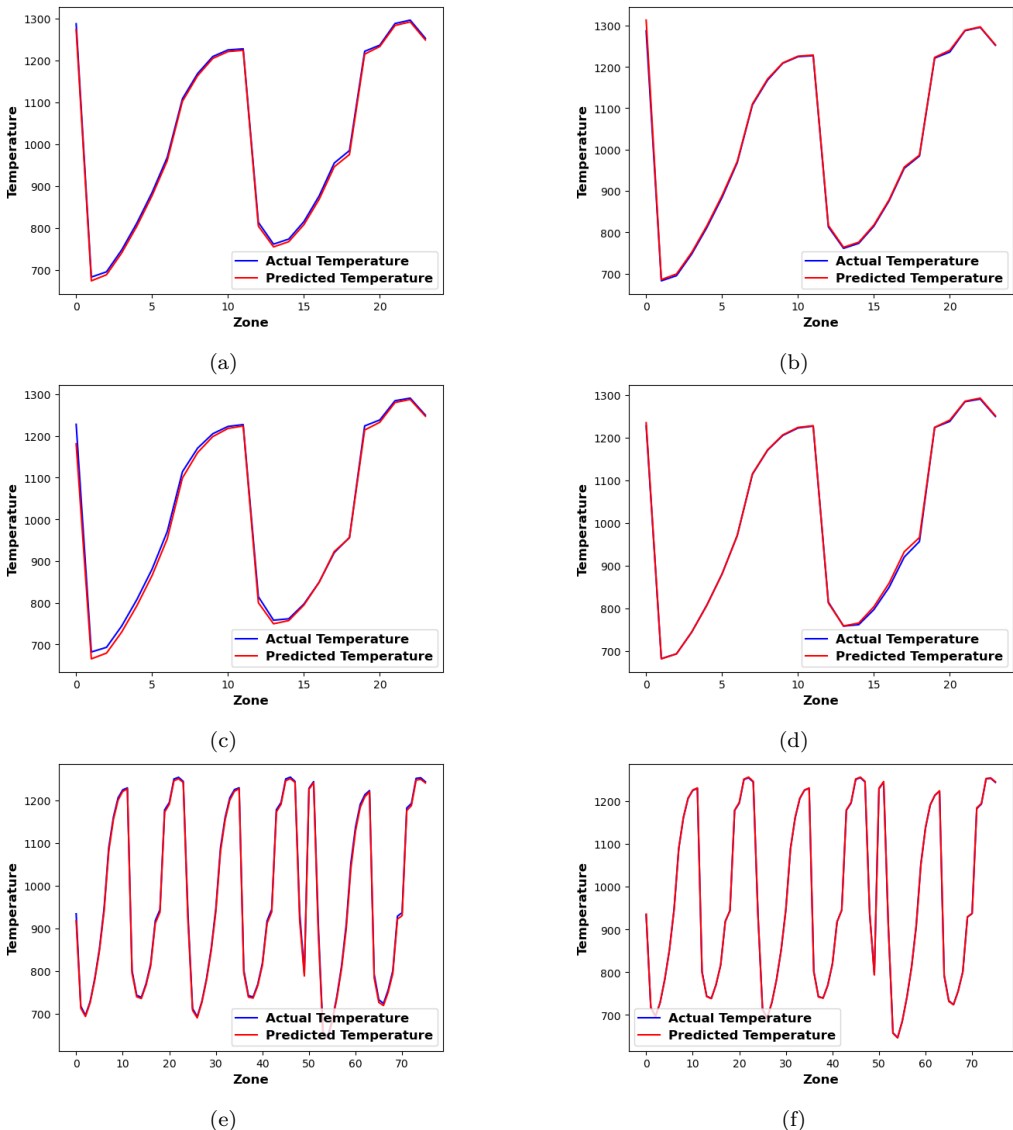

Figure 6: Figure is best viewed in color. Plot of actual (blue) and predicted (red) temperatures (in $^\circ C$) across all volume zones (first two rows) and all furnace surface zones (bottom-most row) using our PINN model EBV+EBS. The left (and resp, right) panels correspond to IID evaluation scenarios without (and resp, with) previous temperatures as inputs. All rows represent predictions for arbitrary time steps. Including previous temperatures in the input for the IID scenario is shown to be beneficial, as also observed empirically. Oscillating pattern observed for both types of zones, with predictions for non-moving/ static furnace surface zones to be relatively better, as per expectations.

performance of the MLP based PINN EBV+EBS is fairly stable across different values, we noticed significant drop in R-MSE tG when $\lambda_{ebv}$ is set to a large value of say, 10. For $\lambda = 0.1$, we noticed reasonable performance throughout, and in particular with respect to R-MSE tS fur. Thus, we set both the values of $\lambda_{ebv}$ and $\lambda_{ebs}$ to 0.1 in our experiments, though, this could vary in different data sets in general, and has to be decided empirically.

We also tried eliminating either of regularization terms. For instance we keep only the EBV term by setting $\lambda_{ebv} = 0.1$ and $\lambda_{ebs} = 0$, and only the EBS term by setting $\lambda_{ebv} = 0$ and $\lambda_{ebs} = 0.1$. But as shown in Table 7, we found that using both instances of volume and surface zone based regularizers together (i.e., $\lambda_{ebv} = 0.1$ and $\lambda_{ebs} = 0.1$) leads to a better performance as compared to either EBV or EBS in isolation.

Table 10: Comparison of our proposed MLP based PINN EBV+EBS method against LSTM, in an IID evaluation setting (on test data). Best result for a row is in bold. ↓: lower metric is better. ↑: higher metric is better.

| Without previous temperatures as inputs | | |
|---|---|---|
| Performance Metric | LSTM | EBV+EBS |
| RMSE tG (↓) | 11.86 | **10.04** |
| RMSE tS fur (↓) | 10.48 | **7.95** |
| RMSE tS obs (↓) | 34.67 | **31.64** |
| MAE tG (↓) | 8.33 | **7.19** |
| MAE tS fur (↓) | 7.15 | **5.58** |
| MAE tS obs (↓) | 16.99 | **15.13** |
| $R^2$ tG (↑) | 0.944 | **0.961** |
| $R^2$ tS fur (↑) | 0.935 | **0.959** |
| $R^2$ tS obs (↑) | 0.848 | **0.885** |
| mMAPE fr (↓) | 11.18 | **6.84** |

In Table 8, we vary the underlying activation functions throughout our model. While we observed the benefits of using ReLU, SiLU, and Mish over others, there is no clear winner. All three lead to competitive performance. But when it comes to consistent performance across batch sizes, we noticed from our experiments that ReLU is more robust. Thus, we could recommend using the basic ReLU as de facto in our experiments.

**Additional comparisons of our MLP based PINN EBV+EBS against classical ML techniques:** In addition to DL, we also compare our EBV+EBS method against the classical ML baselines (in an IID evaluation setting): i) Decision Tree (DT), ii) Random Forest (RF), and iii) Histogram Gradient Boosting (H-GBoost). When it comes to only the classical ML baselines, the performances are as per the expectation. For instance, with previous temperatures as inputs, the performance of DT, RF, and H-GBoost increases. However, being an ensemble learning method, RF performs superior to DT. At the same time, by virtue of boosting, among all the three classical methods, H-GBoost performs the best. We observe superior performance of our model against the classical baselines as well, as reported in Table 9. H-GBoost, though competitive, is significantly slower for our studied case of multi-output regression.

**Additional details and experiments on LSTM:** We now provide additional details about the LSTM variants used in Table 3. The LSTM method has a single LSTM layer with 100 hidden nodes, followed by a FC layer with sigmoid non-linearity. The LSTM2 variant has a single LSTM layer with 50 hidden nodes, followed by FC layer-1 with 50 input nodes and 100 output nodes, FC layer-2 with 100 input nodes and 200 output nodes. Both FC layer-1 and FC layer-2 have ReLU non-linearity. Lastly, there is a final FC layer with sigmoid nonlinearity that maps to the number of output features as in the data set. The DLSTM variant has three stacked LSTM layers, each with 100 hidden nodes, followed by a final FC layer with sigmoid nonlinearity. As we can see, we have kept the total number of layers in LSTM2 and DSLTM similar to that of the baseline MLP.

We noticed that for AR based evaluation, considering different complexities of LSTM variants was useful. However, for the IID evaluation, the performance of more sophisticated LSTM2 and DSLTM were no better than the LSTM method. In Table 10, we report the performance of the LSTM variant in the IID evaluation setting (without previous temperatures as inputs), and showcase that our proposed MLP based PINN EBV+EBS outperforms across the considered performance metrics.

