# OpenReview forum: "Zone Method meets Physics-Informed Neural Networks: Data and Regularization for High-Temperature Processes"
_TMLR — Withdrawn by Authors_

### Review · Reviewer_YtBJ · 2023-12-12

**Summary Of Contributions:**

This paper is about modeling a reheating furnace using a (physics informed) neural network, which obtains best performance. It is a variation of the Zone Method combined with PINNs.

I am not sure what are the contributions of the paper, because the paper makes no claims of contributions, only that the physics informed neural networks works better in standard regression metrics, but this is to be expected and already proven for many applications.

**Audience:**

No

**Claims And Evidence:**

No

**Requested Changes:**

- Please write a proper introduction, with a description of the problem, why the problem is important, and claims and audience statements.
- The claims and audience statements are missing, in particular the paper does not clearly make any claims, and does not clearly state an audience (nor one can be inferred from the paper), so it is not appropriate for evaluation at TMLR.
- Rewrite the whole paper to focus on a machine learning contribution, I believe the authors should clarify what is the contribution to the overall machine learning community from this paper.
- Consider submitting a revised version of this paper to an applications or applied mathematics venue.

**Strengths And Weaknesses:**

Strengths
- The only strength of this paper is to empirically show that a physics informed neural network outperforms a naive baseline and an MLP without physics information in a very physics heavy problem (modeling a reheating furnace).

Weaknesses
- The paper does not have a proper introduction, it goes straight into defining the problem, with an introduction being only three paragraphs at the beginning of the first section. Usually the introduction sets the problem, why it is important, and the contributions/claims of the paper and its audience.
- The paper is mostly mathematical modeling details, there are very little machine learning details, so this makes me feel the paper should be submitted to an applications venue or an applied mathematics journal.
- I believe this paper does not fit the TMLR audience, I truly do not see machine learning researchers being interested in a very niche problem related to rehating furnaces, specially as PINNs being better than stanard neural networks is very obvious, as a standard neural network is not aware of physics constraints.
- The paper does not have ablation results for the regularization coefficients (λ_ebv and λ_ebs), to confirm the effect of each regularization term added to the loss. Here I would have expected tuning the regularization coefficients, as only a single value is used and it not explained how these values were obtained or tuned. The regularization terms are actually very application specific, there is no generalization of those terms to other applications, and I do not see how this method could be generalized to other Machine Learning use cases.
- Overall I believe this is an application paper (not necessarily out of scope for TMLR), but the application is so niche that it is not generalizable to other machine learning tasks or applications.

---

> ### Author Response · Authors · 2024-01-08
> **Response to Reviewer YtBJ**
>
> We would first like to thank you for taking out time to review our paper, and provide valuable comments, which, we believe have helped to polish our draft significantly. We also ended up conducting further experiments, and now have provided more insights to make it easier for a reader to understand our work, and possibly gain some new research ideas. We request you to please go through our revised manuscript, which highlights all the newly added content in red for your ease of reference and consideration of our work. Please note that we have also renamed our paper to "Zone Method meets Physics-Informed Neural Networks: Data and Regularization for High-Temperature Processes" to elaborate our data and loss/regularization specific contributions to the ML audience.
>
> Below, we now try to address each of your comments to the best extent possible.
>
> "paper makes no claims of contributions/ The paper does not have a proper introduction"
> >> We refined our revised draft to propose our claims. We also add a detailed introduction now. Please go through section 1 and 3 of the revised manuscript to better understand this. We apologize for the inconvenience while reading our earlier draft. We thought of experimenting with a different style of writing, but it seems it was not easy to understand. We hope that the revised paper is easier to follow.
>
> "The paper is mostly mathematical modeling details, there are very little machine learning details"
> >> Section 3 (Proposed method) in the revised paper now tries to provide more ML specific details. To supplement it, we make use of Algorithms 2, 3 and 4, and attempt to make it friendlier to the ML audience. We hope that these parts of the revised paper address this concern of yours.
>
> "I believe this paper does not fit the TMLR audience"
> >> Can you please reconsider our revised manuscript, to see if it helps the ML reader better ? We now have added further experiments with LSTMs, and have also shown that the proposed PINN framework could encompass various architectures generically, e.g., MLP, LSTMs, to come up with newer physics-based methods with existing regression objectives.
>
> "PINNs being better than standard neural networks is very obvious, as a standard neural network is not aware of physics constraints"
> >> While this is true in theory, like with all other ML techniques, we need to empirically assess and establish this in practice. And the novelty part comes into the picture with each new problem using PINN is being addressed, and this is not trivial.
>
> "paper does not have ablation results for the regularization coefficients"
> >> "In-depth analysis of MLP based PINN EBV+EBS:" paragraph of appendix, along with Fig 5, and Table 7 addresses this.
>
> "Please write a proper introduction, with a description of the problem, why the problem is important, and claims and audience statements./ claims and audience statements are missing,"
> >> Kindly go through section 1 of the revised paper.
>
> "Rewrite the whole paper to focus on a machine learning contribution,"
> >> We have rewritten the entire paper now, while also renaming it to better suit the ML audience.

---

### Review · Reviewer_iopX · 2023-12-23

**Summary Of Contributions:**

The paper proposes a Physics-Inspired Neural Network (PINN) for training an MLP for predicting furnace parameters related to temperature and firing rate for a given furnace system at a given time. The paper first outlines the problem formulation after a brief introduction, including relevant input and output variables for the PINN as well as the loss formulation that includes physics-based regularization terms. The paper then describes the formulation of the loss terms and presents a series of experiments related to PINN training and compare the proposed PINN to a naive guess and a regular MLP. The experiments generally show that the PINN performs better than the simple baselines. Next, the author provide some discussion and a conclusion.

**Audience:**

No

**Claims And Evidence:**

No

**Requested Changes:**

Please address the weaknesses identified in the section above. I think that a related work section and additional experiments to assess the performance of the method compared to relevant baselines and OOD generalization would significantly strengthen the paper.

**Strengths And Weaknesses:**

Strengths:
- The paper studies a relevant problem for industrial applications of machine learning.

Weakness:
- No related work section on PINNs. It would be good ton know how the proposed method, especially the loss terms which seem to be the primary contribution, relate to previous work.
- The set of tested experiments and the problem setting tackled is quite basic, both in the design of the method and the underlying problem. The authors mention adding sequence-based methods, such as RNNs, at the end of the paper. I think including such models would be required to make to the paper interesting for the TMLR audience.
- The paper does not outline the relevant compute costs and potential savings from their proposed method. For example, how long does it take an accurate simulator to obtain the results compared to the proposed PINN? This is both for the data generation method and more sophisticated simulation methods, such as CFD. The authors should explain which methods are most appropriate for their setting.
- The chosen baselines are also quite basic. I think the paper needs a more thorough review of appropriate baselines and potential implementation of them.
- The OOD setting is not studied in the current version of the paper. Based on the authors' own writing in the introduction, this is the most important setting and should be studied in a broader set of experiments.
- The data generation is only described in the appendix. I recommend moving it and adding more relevant discussion and description to the main text.

---

> ### Author Response · Authors · 2024-01-08
> **Response to Reviewer iopX**
>
> We would first like to thank you for taking out time to review our paper, and provide valuable comments, which, we believe have helped to polish our draft significantly. We also ended up conducting further experiments, and now have provided more insights to make it easier for a reader to understand our work, and possibly gain some new research ideas. We request you to please go through our revised manuscript, which highlights all the newly added content in red for your ease of reference and consideration of our work. Please note that we have also renamed our paper to "Zone Method meets Physics-Informed Neural Networks: Data and Regularization for High-Temperature Processes" to elaborate our data and loss/regularization specific contributions to the ML audience.
>
> Below, we now try to address each of your comments to the best extent possible.
>
> "No related work section on PINNs / especially the loss terms which seem to be the primary contribution, relate to previous work"
> >> We have now rectified this in the revised draft. Our first draft was an experiment to write the paper in a non-conventional style, and we believe it might have caused inconvenience to the reviewers to understand our work. We apologize for this. Thus, we add a detailed Related work section, along with a better introduction, to elaborately describe our work, and how it uniquely places itself among the body of literature. We also have renamed our paper now to focus on our core ML-specific contributions.
>
> "authors mention adding sequence-based methods, such as RNNs, at the end of the paper. I think including such models would be required to make to the paper interesting for the TMLR audience."
> >> In our revised paper, we now not only report comparisons against LSTMs, but also showcase that our physics-based regularizers could also be used along with them to further improve their performance. We now, overall, showcase Physics-inspired/-based methods corresponding to both MLP and recurrent architectures. Our detailed Algorithms also provide explicit steps to implement our work.
>
> "paper does not outline the relevant compute costs and potential savings from their proposed method."
> >> Please refer to newly added section 4.4 of the revised paper for details on this.
>
> "chosen baselines are also quite basic"
> >> We understand this skepticism. However, the classical baselines, though simple, are widely adopted standards for the real-world industry for commercial usage, as they are interpretable, and perform well (under fair comparisons against sophisticated methods in literature for tabular/IID data). Nevertheless, for this particular use-case, we further provide experiments on different recurrent variants based on LSTM, and also propose further novel physics-based LSTM variants using our novel regularizers.
>
> "OOD setting is not studied"
> >> We did create our data set to be inherently OOD in nature. More details are provided in the main text (section 3.2.1) as well as the appendix, in our revised paper.
>
> "The data generation is only described in the appendix. I recommend moving it and adding more relevant discussion and description to the main text."
> >> Done. Kindly refer to Section 3.2 of the revised paper, and please let us know if this reads well now.

---

> > ### Comment · Reviewer_iopX · 2024-01-15
> > **Major Changes in the Draft Require New Rounds of Reviews**
> >
> > Thank you for the extensive edits. While I think they go to great lengths to making the paper and addressing some of the concerns raised by myself and other reviewers, I believe that the edits made by the authors change the paper so significantly that a new round of reviews are required. Additionally, the current draft exceeds the recommend page limit for the original review considerations.

---

> > > ### Comment · Reviewer_YtBJ · 2024-01-15
> > > **Agree with New Round of Reviews**
> > >
> > > Thanks for this comment, as a reviewer I also agree that the paper requires a new round of reviews, content has changed so much that it is almost like a new paper, this has delayed my final review and recommendation as I basically have to read and analyze the whole paper again.

---

### Review · Reviewer_fUtj · 2023-12-30

**Summary Of Contributions:**

The authors propose a novel loss function for the PINN architecture to predict the reheating furnace temperature profile. The proposed loss function consists of items for the volume and surface zones. They are computed based on the physical properties of the furnace and heat transfer coefficients. The presented numerical tests show the performance of the proposed scheme compared to MLP architecture and naive averaging for some of the considered quantities.

**Audience:**

Yes

**Broader Impact Concerns:**

No concerns about the ethical implications of the work.

**Claims And Evidence:**

No

**Requested Changes:**

1) please carefully use \cite{} and \citep{} commands to increase the readability of the text
2) the key assumption of PINNs is the knowledge of the underlying PDEs corresponding to the observed data and backpropagation through them. So, please add explicitly such PDEs in the text and provide more details on how one can use backpropagation in the considered setup
3) the intuition behind the introduced regularizers is unclear for non-experts. Please provide clearer motivation and improve the readability of this Section since it is the core of the manuscript
4) what does DFA mean?
5) what do you mean by "PINN against a baseline MLP with the same architecture as our PINN"? What ingredient of PINN has the same architecture and how does it affect the overall workflow?
6) please add remarks on whether train or test losses are reported in Tables 1 and 2. And how the test set is generated?
7) the conclusion section looks like the subpart of the related work section. Please revise it and include a summary of the presented work, main results, and possible future work
8) please consider alternative existing regularizers to train PINN, e.g. from this work https://arxiv.org/pdf/2110.09813.pdf
9) please discuss how to select the coefficients for the introduced loss ingredients

**Strengths And Weaknesses:**

**Strengths**

The paper considers a specific modeling problem and proposes a modification of the loss function for a PINN model. The authors show the performance of the proposed PINN model equipped with the loss function compared to MLP and naive averaging. The presented experimental results show that the proposed approach is promising for accurate modeling of the process inside the reheating furnaces.

**Weaknesses**

The main weakness of the presented manuscript is focusing on the reheating furnace modeling problem rather than on the proposed modification of the PINN loss. Therefore, the overall readability of the manuscript can be significantly improved to explain the novelty and analyze it extensively. The current version of the manuscript lacks the mathematical description of the underlying PDE (Algorithm 1 includes a lot of undefined notations like DFA), particular features of backpropagation, and comparison with reasonable alternatives from the PINN-type models.

---

> ### Author Response · Authors · 2024-01-08
> **Response to Reviewer fUtj**
>
> We would first like to thank you for taking out time to review our paper, and provide valuable comments, which, we believe have helped to polish our draft significantly. We also ended up conducting further experiments, and now have provided more insights to make it easier for a reader to understand our work, and possibly gain some new research ideas. We request you to please go through our revised manuscript, which highlights all the newly added content in red for your ease of reference and consideration of our work. Please note that we have also renamed our paper to "Zone Method meets Physics-Informed Neural Networks: Data and Regularization for High-Temperature Processes" to elaborate our data and loss/regularization specific contributions to the ML audience.
>
> Below, we now try to address each of your comments to the best extent possible.
>
> Strength + Weakness: "proposes a modification of the loss function for a PINN model" / main weakness: "focusing on the reheating furnace modeling problem rather than on the proposed modification of the PINN loss."
> >> Thanks for acknowledging this. In our revised paper, we put further emphasis on it, and highlight it better.
>
> "manuscript lacks the mathematical description of ..."
> >> We put more details of each of the terminologies introduced in our revised paper. Specifically, we request you to kindly go through our revised appendix to better understand all the details of the governing physics, especially, in terms of "Exchange Factors, TEA, DFA, and WSGG" (kindly note that, in this very unique case, there is no PDE involved, but rather, we have a "system of simultaneous equations consisting of concepts like TEA/DFA, WSGG, etc". Thus, we describe them in the appendix).
>
> Requested Changes:
> 1. carefully use \cite{} and \citep{}
> >> We have now taken care to rectify this. Please help us understand if the draft needs further improvements.
>
> 2. please add explicitly such PDEs in the text and provide more details on how one can use backpropagation in the considered setup
> >> As described above, we add more details on the governing physics by virtue of Exchange Factors, TEA, DFA, and WSGG in the Appendix. The newly introduced Algorithm 2 in the revised paper helps the reader understand implementation details of the backpropagation in detailed pytorch-style code. This Algorithm is further supplemented with Algorithms 3 and 4 providing details of all the helper functions used, in terms of code.
>
> 3. intuition behind the introduced regularizers is unclear for non-experts.
> >> We hope our discussion on "Exchange Factors, TEA, DFA, and WSGG" in Appendix, and inclusion of more text in the main body (e.g., "background" paragraph in the "proposed method" section of revised paper) has perhaps addressed this ?
>
> 4. what does DFA mean?
> >> point 3 above also covers this.
>
> 5. what do you mean by "PINN against a baseline MLP with the same architecture as our PINN"?
> >> We feel that it should be clearer in the revised paper now. However, for your convenience, we would like to mention here that the architecture of both are the same. The difference comes in the objective function. Baseline MLP has only the L_sup term, but the proposed MLP-based PINN EBV+EBS has two additional regularizer terms L_ebv and L_ebs, respectively taking care of the EBV and EBS equations inherently. These two additional terms make the resulting network physics-aware.
>
> 6. please add remarks on whether train or test losses are reported in Tables 1 and 2. And how the test set is generated?
> >> We now have explicitly added this in each of the tables in the paper, although it was already mentioned in the main body earlier. Detailed description of data set generation is now significantly elaborated in the main body in section 3.2.1 as well as the appendix.
>
> 7. conclusion section looks like the subpart of the related work
> >> We took care of it now.
>
> 8. please consider alternative existing regularizers to train PINN, e.g. from this work https://arxiv.org/pdf/2110.09813.pdf
> >> We thank you for the interesting pointer. However, the work pointed is just a way to automatically tune hyperparameters for the regularization terms. In our work, we could already get desired results by empirical tuning of hyperparameters, and thus, we believe that this exercise may not be strongly beneficial to our work.
>
> 9. please discuss how to select the coefficients for the introduced loss ingredients
> >> "In-depth analysis of MLP based PINN EBV+EBS:" paragraph of appendix, along with Fig 5, and Table 7 addresses this.

---

> > ### Comment · Reviewer_fUtj · 2024-02-04
> > **Response to the revision**
> >
> > Thanks for the significant revision of the submitted manuscript and for taking into account my comments and requests. Since most parts of the manuscript were rewritten, I agree with other reviewers that a new round of reviews is needed for proper evaluation of the almost new text of this manuscript.
> > The authors added a lot of new experimental results and theoretical explanations of the introduced regularizers. These updates and changing the title of the paper may be considered as a new submission rather than a revision of the original manuscript.

---

### Author Response · Authors · 2024-01-08
**Highlights of the revised paper to address the consolidated reviewer comments**

We would first like to thank all the reviewers for taking out their time to review our paper, and provide valuable comments based on their expertise, which, we believe, have helped to polish our draft significantly.

We also thank the action editor for giving us the opportunity to update our work with a revision, and respond to the reviewer comments.

We also ended up conducting further experiments, and now have provided more insights to make it easier for a ML reader to understand our work, and possibly gain some new research ideas. We request you all to please go through our revised manuscript, which highlights all the newly added content in red for your ease of reference and consideration of our work. Please note that we have also renamed our paper to "Zone Method meets Physics-Informed Neural Networks: Data and Regularization for High-Temperature Processes" to elaborate our data and loss/regularization specific contributions to the ML audience.

Below, based on reviewer suggestions, are the key modifications made in the revised manuscript:
1. Renaming of the paper to shift focus to ML aspects.
2. Major rewriting of the entire paper, with specific focus on Abstract, Conclusions, Introduction (with explicit claims/ contributions), Related work (detailed placement of our work with the existing "similar sounding" literature).
3. Following are our key contributions: i) Data generation for the problem, first temporal, and then IID reconstruction. Inherently, data is generated to provide a testbed for evaluating OOD generalizability. ii) Generic PINN framework, which can easily be applied to a standard regression model (MLP, LSTMs, etc,). iii) Detailed experiments, along with qualitative and quantitative study in-depth. iv) Code-snippets provided to make the reader understand implementation details, and reproduce in their own use-cases.
4. Provision of Background in the proposed method section to further elaborate details about governing physics (along with supplemental details in the appendix).
5. Detailed data generation explained algorithmically (Algorithm 1) as well as using visual block diagram (Fig 2). Further text based details also added in the corresponding section 3.2.
6. Algorithms 2, 3, and 4 provide implementation details in pytorch-styled code, which should be helpful to understand the backpropagation.
7. Throughout the text we add subsections to make the paper easier to read.
8. Section 4.3 also showcases visual plots of temperature predictions.
9. Section 4.4 discussed computational aspects.
10. Section 4.5 reports results with LSTMs, and Section 4.6 discusses how physics-based LSTMs can be obtained to achieve even better results than the already best performing proposed MLP-based PINN called EBV+EBS.
11. Appendix now has additional content: details of "Exchange Factors, TEA, DFA, and WSGG", helper functions for the core Algorithm implementation in pytorch-style, visual plots (Fig 5) involved in the decision-making process of tuning of hyperparameters/coefficients for the physics based regularization terms, further visual plots of temperature predictions, Additional details and experiments on LSTM.

We hope that the above details provide the ML reader a better opportunity to understand our work, and hopefully give at least some relevant research insights.

---

### Note · Authors · 2024-02-04

**Comment:**

Dear AE,

We are withdrawing our manuscript due to concerns about the review process. We appreciated the reviewers' insights but found their comments sought more clarity than addressing technical issues. Despite our prompt revision, the reviewers suggested treating our work as a new submission. This, coupled with the delay in receiving the initial reviews (based on https://jmlr.org/tmlr/reviewer-guide.html), has led us to seek publication elsewhere. We hope our experience can help improve the review process.

Sincerely,

**Withdrawal Confirmation:**

I have read and agree with the venue's withdrawal policy on behalf of myself and my co-authors.